# Untargeted pixel-by-pixel metabolite ratio imaging as a novel tool for biomedical discovery in mass spectrometry imaging

Huiyong Cheng[1], Dawson Miller[1], Nneka Southwell[2], Paola Porcari[3], Joshua L Fischer[4], Isobel Taylor[1], J Michael Salbaum[5], Claudia Kappen[5], Fenghua Hu[6], Cha Yang[6], Kayvan R Keshari[3], Steven S Gross[1], Marilena D'Aurelio[2], Qiuying Chen[1]*

[1]Department of Pharmacology, Weill Cornell Medicine, New York, United States; [2]Brain and Mind Research Institute, Weill Cornell Medicine, New York City, United States; [3]Memorial Sloan Kettering Cancer Center, New York, United States; [4]Bruker Daltonics, Billerica, United States; [5]Pennington Biomedical Research Center, Louisiana State University, Baton Rouge, United States; [6]Cornell University, Department of Molecular Biology & Genetics, Ithaca, United States

## eLife Assessment

This **valuable** study describes a software package in R for visualizing metabolite ratio pairs. The evidence supporting the claims of the authors is **solid** and broadly supports the authors' conclusions. This work would be of interest to the mass spectrometry community.

*For correspondence: qic2005@med.cornell.edu

**Abstract** Mass spectrometry imaging (MSI) is a powerful technology used to define the spatial distribution and relative abundance of metabolites across tissue cryosections. While software packages exist for pixel-by-pixel individual metabolite and limited target pairs of ratio imaging, the research community lacks an easy computing and application tool that images any metabolite abundance ratio pairs. Importantly, recognition of correlated metabolite pairs may contribute to the discovery of unanticipated molecules in shared metabolic pathways. Here, we describe the development and implementation of an untargeted R package workflow for pixel-by-pixel ratio imaging of all metabolites detected in an MSI experiment. Considering untargeted MSI studies of murine brain and embryogenesis, we demonstrate that ratio imaging minimizes systematic data variation introduced by sample handling, markedly enhances spatial image contrast, and reveals previously unrecognized metabotype-distinct tissue regions. Furthermore, ratio imaging facilitates identification of novel regional biomarkers and provides anatomical information regarding spatial distribution of metabolite-linked biochemical pathways. The algorithm described herein is applicable to any MSI dataset containing spatial information for metabolites, peptides or proteins, offering a potent hypothesis generation tool to enhance knowledge obtained from current spatial metabolite profiling technologies.

## Introduction

Recent developments in mass spectrometry imaging (MSI) enable spatial mapping of the relative distribution and abundances of proteins, peptides, lipids, small molecule metabolites, drugs and elemental isotopes in heterogeneous tissue sections (*Shimma, 2022*) with resolution at, or near, cellular (*Zhang et al., 2023a*). Hence, MSI has become a promising tool in the era of spatial-omics (*Zhang et al., 2023b*; *Wang et al., 2023*; *Soudah et al., 2023*; *Sivanesan et al., 2023*; *Rončević et al., 2023*; *Moore et al., 2023*; *Liu et al., 2023*; *Li et al., 2023*; *Jiang et al., 2023*; *Ikegawa et al., 2023*; *Gorman et al., 2024*; *Gitta et al., 2023*; *Zou et al., 2022*), especially at the single cell level (*Zhao et al., 2023*). Matrix-assisted laser desorption and ionization (MALDI), secondary ion mass spectrometry (SIMS), and desorption electrospray ionization (DESI) are the dominant modes of ionization for MSI. Regardless of ionization method, pixel size, and knowledge of structural identity, all MSI analyses provide pixel-by-pixel relative abundances of all detected charged molecular masses. Current MSI data processing tools provide 2D or 3D image data visualization as heatmaps from normalized pixel-by-pixel data of acquired mass spectra (*Vos et al., 2021*; *Verbeeck et al., 2020*; *Alexandrov, 2012*; *Trede et al., 2012*) and are mainly limited to relative abundance data only.

Metabolite ratio imaging has been used in medical MRI spectroscopy to study metabolite distribution within tissue (*Riches et al., 2009*). SIMS ratio imaging has been used in biological, material, and environmental sciences (*Jia et al., 2023*; *Pumphrey et al., 2009*). Additionally, targeted serum and urine metabolite ratios have aided assessment of genome-wide GWAS and MWAS association of metabolic traits by serving as proxies for enzymatic reaction rates (*Suhre et al., 2011*; *Illig et al., 2010*; *Gieger et al., 2008*). Hypothesis-free testing of ratios between all possible metabolite pairs in tissue extracts analyzed by GWAS and MWAS have provided an innovative approach for discovery of new biologically meaningful molecular associations (*Petersen et al., 2012*). Furthermore, metabolite ratios have been used to construct large-scale neural networks that improve statistical assessments and facilitate data interpretation (*Lassen et al., 2023*). In geographical and hyperspectral imaging, several free and commercially available software packages are available for band ratio computation and imaging. These include: (1) ENVI IDL, a geospatial imaging tool that allows ratio computation between spectral bands (https://www.nv5geospatialsoftware.com/Products/ENVI); (2) MATLAB image processing toolbox for hyperspectral imaging (https://www.mathworks.com/matlabcentral/fileexchange/50340-ratioimage); (3) Spectral Python package (Spy, https://www.spectralpython.net/); and (4) QGIS with plugins can be used for hyperspectral image analysis with a ratio between bands (https://www.qgis.org/en/site/). The above four tools can perform an individual ratio image from an input pair, but may require additional programing to image ratios from the untargeted metabolite pairs in an MSI dataset (*Supplementary file 1*).

To fill this gap, we developed an untargeted computational R workflow to image ratios of all detected metabolites in every pixel of an MSI experiment. While individual ion abundances in spatial metabolite profiling fail to inform on metabolic pathway activity, or on the identity of metabolic intermediates contributing to these pathways, pixel-by-pixel imaging of the ratio of an enzyme's substrate to its derived product may offer an opportunity to view the distribution of functional activity for a given metabolic pathway across tissue. In this report, we use MALDI MSI data obtained from cryosections of murine embryos, brains, and a mitochondrial myopathy model of tissue derangements to demonstrate the application of ratio imaging to recognize both normal and pathology-associated regiospecific metabolic perturbations. We further show that ratio imaging minimizes systematic variations in MSI data introduced by sample handling, improves image resolution, enables anatomical mapping of metabotype heterogeneity, facilitates biomarker discovery, and reveals new spatially resolved tissue regions of interest (ROIs) that are metabolically distinct but otherwise unrecognized. Using murine embryo and hippocampus MALDI MSI data acquired at single-cell spatial resolution, we showcase the potential of single cell metabolite ratios to probe metabolic enzyme activities, as well as the potential for integration with other datasets obtained using multiomic data acquisition platforms. Importantly, the algorithm for ratio imaging described should be applicable to spatial MS profiling of all sorts, including metabolites, peptides, and proteins. This software package is being made available as a potent tool to enhance knowledge obtained from conventional spatial metabolite profiling analyses (https://github.com/qic2005/Untargeted-mass-spectrometry-ratio-imaging copy archived at *qic2005, 2024*).

# Materials and methods

**Key resources table**

| Reagent type (species) or resource | Designation | Source or reference | Identifiers | Additional information |
|---|---|---|---|---|
| Biological sample (*Mus musculus*) | Mouse brain | Animal facility Weill Cornell Medicine | n/a | Fresh frozen |
| Biological sample (*Mus musculus*) | Mouse adipose | Animal facility Weill Cornell Medicine | n/a | Fresh frozen |
| Chemical compound, drug | N-(1-Naphthyl)ethylenediamine dihydrochloride | Sigma-Aldrich | Cat # 222488 | |
| Chemical compound, drug | 9-Aminoacridine | Millipore Sigma | Cat # 92817 | |
| Other | Indium tin oxide | ITO; (Delta Technologies) | Cat # CB-90IN-S111 | |
| Chemical compound, drug | 1,5-Diaminonaphthalene | Millipore Sigma | Cat # 56451 | |

## Sample preparation

Brain and adipose tissues from a mitochondrial myopathy model of cytochrome c oxidase assembly factor heme A:farnesyltransferase COX10 knockout (COX10 KO) and wildtype (WT) mice were frozen in liquid nitrogen and stored at −80 °C until processing. Entire mouse decidua dissected from pregnant females at gestation day 8.5 were embedded in 2% carboxymethylcellulose, frozen in liquid nitrogen, stored at −80 °C, and sectioned at 12 μm nominal thickness. Brain cryosections (n=4 per group) were cut at 10 μm and adipose at 20 μm thickness (n=3 for wildtype control and n=4 for COX 10 KO), mounted on conductive slides coated with indium tin oxide (ITO; Delta Technologies; cat # CB-90IN-S111) and stored at −80 °C. On the day of MALDI MS data acquisition, ITO-slides with tissue sections were transferred to a vacuum chamber and dried for 30 min prior to deposition of desired matrices for imaging: N-(1-naphthyl)ethylenediamine dihydrochloride (NEDC; 10 mg/ml in 75% methanol) for murine brain and E8.5 embryos; 1,5-diaminonaphthalene (DAN, 2 mg/mL in 50% acetonitrile) for murine COX10 KO adipose tissues, and 9-aminoacridine (9AA, 5 mg/ml in 85% Ethanol) for murine hippocampus. All matrices were delivered using an HTX TM-Sprayer (HTX Technologies LLC, NC) with optimized spraying parameters for each individual matrix. Matrix-coated tissue sections were dried in vacuum desiccator for 20 min before MALDI MSI data acquisition in negative ion detection mode.

## MALDI MSI data acquisition and processing

MALDI MSI data were acquired at raster width of either 10 μm or 80 μm using a 7T scimaX-MRMS mass spectrometer (Bruker Daltonics, USA) equipped with a SmartBeam II laser and a MALDI source. Peak-picked MALDI-IMS data were imported into SCiLS Lab 2024a software (SCiLS, Bremen, Germany) for image visualization. Compound identifications were assigned based on both accurate mass (<2 ppm mass accuracy) and isotope pattern matches to free source metabolite and lipid databases, including the Human Metabolome Database (HMDB), KEGG and LIPIDMAPs. An in-house R-code developed for the programmable SCiLS Lab Application Interface (SCiLS lab API) was used for fast export of raw pixel-by-pixel metabolite and unknown mass spectral abundance data from ROIs and whole tissue sections. In commercial processing software where missing values either have no value or have solid zero in abundance, we need to annotate the missing value for ratio calculation. Towards this, we first obtain the minimum abundance of a particular m/z among all pixels with detectable abundance (i.e. excluding missing values), then use 1/5 this minimum value as a threshold to annotate missing value. Pixel-by-pixel metabolite ratios between any two detected metabolites/features were calculated by applying R combination and ratio function to the annotated pixel data (see R code). Differential metabolite ratios among ROIs were visualized in R and exported as either an image or.PDF file. *Figure 1* depicts a schematic workflow using SCiLS Lab API for metabolite ratio imaging and associated multivariate analysis among ROIs. Notably, a series of SCiLS lab API R codes for ratio imaging visualization, data analysis, and UMAP segmentation are available for download on GitHub (https://github.com/qic2005/Untargeted-mass-spectrometry-ratio-imaging copy archived at *qic2005, 2024*). For MSI data acquired from other non-Bruker MS imaging instruments, R codes are also available to

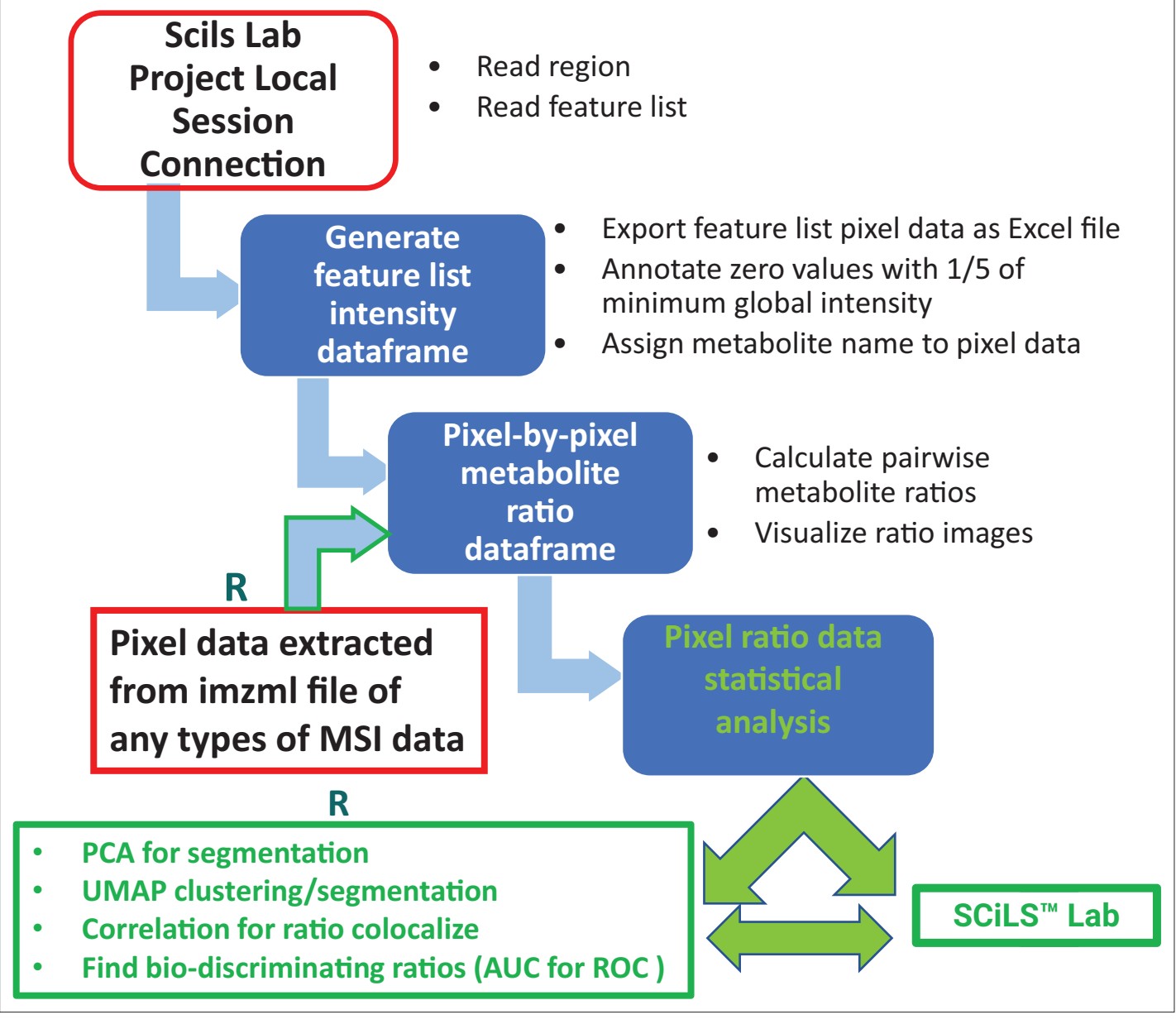

**Figure 1.** Schematic workflow of metabolite ratio imaging using pixel data from Bruker SCiLS Lab API or imzml file from any MSI data source. For SCiLS lab MSI data, the execution of R codes starts with connecting a local SCiLS lab session project in R studio. Installation of SCiLS lab is required for this connection. After pixel data for a feature list is generated and exported as an Excel csv file, SCiLS lab installation is not required for downstream ratio imaging and data analysis, except for the potential writing of a spot image with labels back to SCiLS lab. For other types of MSI data, R code execution extracts pixel data from imzml file for subsequent ratio imaging and analysis.

The online version of this article includes the following source data and figure supplement(s) for figure 1:

**Figure supplement 1.** Application of pixel-by-pixel exemplary MALDI MS imaging data to metabolomics multivariate analysis using free or commercially available software package.

**Figure supplement 2.** Exemplary MALDI MSI images generated before (**A**) and after (**B**) missing value annotation and from reciprocal ratios (**C–D**).

**Figure supplement 2—source data 1.** Source data contains pixel data used to plot the figure.

**Figure supplement 3.** Ratio images obtained from metabolite ion abundance exhibit similar characteristics to those generated from absolute metabolite concentration.

**Figure supplement 3—source data 1.** Source data contains pixel data for this figure.

extract pixel data directly from.imzml file, a common data format for MS imaging, using Cardinal 3.0 free-source software and the resulting pixel data can be applied for ratio imaging (*Figure 1*).

## Results

### Generation of an untargeted spectral feature list for export of pixel data to enable downstream metabolite ratio analysis

Using SCiLS lab API, we sought to establish new data processing and analysis features for use in metabolite ratio imaging. To this end, we developed an R code that engenders fast export of raw or normalized pixel-by-pixel feature abundance data for all sections and/or regions of interest (ROIs) in an Excel.csv file format. A targeted feature list from metabolic pathways of interest can also be imported into SCiLS lab as.mir files, generated using either a Python program or a SCiLS lab feature list with manual ID annotation of peak-picked spectral data. Feature lists from multiple pathways of interest can be saved for exclusive ratio imaging of these designated pathways. Once pixel data are available, a SCiLS lab license is no longer required for metabolite ratio imaging and downstream statistical analysis, except for label and spot image generation within the SCiLS lab environment. Notably, Imaging Mass Spectrometry Markup Language (imzml) is a common data format for MSI. It was developed to allow the flexible and efficient exchange of large MS imaging data between different instruments and data analysis software (*Schramm et al., 2012*). It contains two sets of data: the mass spectral data which is stored in a binary file (.ibd file) to ensure efficient storage and the XML metadata (.imzml file) which stores instrumental parameters, sample details. Therefore, imzml file can not be directly used. We also developed codes to generate Excel format pixel data from.imzml files of other vendor's proprietary MSI software are also amenable to the ratio imaging strategy described herein. Therefore, we include a R code for extracting pixel data directly from imzml file, regardless of source of datafile from different vendors. Notably, pixel-by-pixel metabolite imaging data can also be co-registered with other omics imaging datasets obtained from a sequential serial cryosection, independent of the need for a SCiLS lab license.

Given the lack of need for proprietary software to export pixel-by-pixel data, an individual feature list provides MALDI-MS instrument/application flexibility. Additionally, considering each pixel as a replicate from a defined ROI, exported pixel data can be readily used with either vendor-specific or non-proprietary metabolomics data software platforms for biomarker discovery and multivariate spatial metabolomics analysis such as principal component analysis (PCA), partial least-squares discriminant analysis (PLS-DA), hierarchical clustering (HCA), ROC curve assessment, VIP scoring, and pathway enrichment (*Pang et al., 2022*; *Thompson and Moseley, 2023*; *Powell and Moseley, 2023*; *Sud et al., 2016*; *Rodeiro et al., 2023*; *Du et al., 2023*; *Dekermanjian et al., 2021*; *Plyushchenko et al., 2022*; *Misra, 2021*; *Chang et al., 2021*; *O'Shea and Misra, 2020*; *Adams et al., 2020*; *Misra, 2020*). *Figure 1—figure supplement 1* shows an exemplary conversion of ROI pixel data to formats compatible with import into MetaboAnlayst (*Pang et al., 2022*) and Agilent Mass Hunter Professional. For the presented data, brain outer cortex regions were selected for comparison between mitochondrial cytochrome oxidase 10 (COXKO) and WT (n=4 for each genotype) mice.

### MSI ratio imaging requires missing pixel abundance annotation but not absolute metabolite quantification

Missing value annotation is conventionally employed for interpretation of omics data, including metabolomics (*Shahjaman et al., 2021*; *Wei et al., 2018*; *Jin et al., 2018*; *Di Guida et al., 2016*). Spatial metabolomics data acquired from MALDI and DESI MSI studies are subjected to missing value annotation when features are undetected in a given pixel. Given the differences in cell types and ionization efficiency/suppression across tissue regions, missing values are more common in spatial MSI than LC/MS data from the same tissue, especially in the case of FT-ICR-MS, which uses a peak-picking threshold to filter out the bulk of signal noise generated from Fourier data transformation. To prevent potential inclusion of a missing value as 'zero' abundance for imaging data analyses that include ROI metabolite mean abundance comparisons, we annotate missing values with 1/5 the minimum value quantified in all pixels in which it was detected. *Figure 1—figure supplement 2A-B* show the indistinguishable images of exemplary metabolites before and after missing value annotation (*Pang et al., 2022*). Therefore, pixel-wise missing value annotated data was used for all statistical analyses,

image plots, and metabolite ratio imaging described herein. To provide reciprocal and complementary images for easier ROI access and visualization, we consider images for both A/B and B/A ratios (*Figure 1—figure supplement 2C-D*).

It is possible that differences in ionization efficiency among metabolites may result in differential images obtained from ion abundance ratio compared to concentration ratio. To test this, we quantify the absolute concentration of brain glucose, lactate, and ascorbate using stable isotope standards spiked into the mimetic tissue sections prepared from carboxymethylcellulose (CMC) embedded brain homogenate. *Figure 1—figure supplement 3* shows similar ratio images among lactate, glucose, and ascorbate obtained from abundance data compared to quantified concentration data. Although stable isotope standards and mimetic tissue model are often used to obtain quantitative concentration of metabolite/lipid of interest, it is not applicable for untargeted metabolite ratios that include an assessment of structurally undefined species. Note that the utilization of our strategy is to provide untargeted (and targeted) ratio imaging as a hypothesis generation tool and our quantification data indicate this use does not require absolute metabolite quantification.

## Pixel-by-pixel ratio imaging reduces cryosection preparation artefacts and improves data interpretation

Preparing cryosections from non-fixed and non-embedded tissues is a technically demanding procedure prone to sectioning artefacts (*Rieppo et al., 2004*). Tissue freezing, cryo-cut temperature variation, section thickness irregularities, and tissue storage can all affect imaging performance. Pixel-by-pixel ratioing of metabolite/feature pairs offers the important advantage of reducing artefacts caused by non-uniform matrix coating of cryosections and other elements of sample processing, especially for slides containing multiple tissue cryosections. This is because as tissue cryosections are thaw-melted onto slides individually, the layout sequence results in variable waiting times for completion of the mounting process. Thus, individual cryosections undergo varying degrees of metabolically-active metabolite degradation, even at the typical –20 °C cryotome operating temperature.

*Figure 2A–C* and *Figure 1—figure supplement 2B* compare images for lysophosphatidylethanolamines (LPEs 18:1, 20:4, 22:4, 22:6) on brain sections from COX10 KO and WT mice. Lysophospholipids contain a free hydroxyl moiety in either the sn-1 or sn-2 position of the glycerol backbone and are generated by phospholipase-mediated (PLA1 and PLA2, respectively) hydrolysis of phospholipids. PLA1 hydrolyzes an ester bond at the sn-1 position, producing a saturated or monounsaturated fatty acid and a 1-lyso-2-acyl-phospholipid, while PLA2 preferentially hydrolyzes unsaturated fatty acids at the sn2 position, producing polyunsaturated fatty acids and a 2-lyso-1-acyl-phospholipid.

Notably, LPEs with polyunsaturated acyl chains of 20:4, 22:4, and 22:6 shown in *Figure 2* predominantly originate from PLA1 action. As shown, images of these individual LPEs are relatively diffuse and variable among replicates of the same genotype (*Figure 2A–C*). A relatively higher abundance of these species was imaged in cross-sections of the striatal ventral region of the brain, except for LPE 22:6 in COX10 KO brain, which was more abundant in the outer cortex. Note that 20:4 and 22:4 acyl chains are derived from the $\omega 6$ fatty acids arachidonic acid (AA) and adrenic acid, whereas 22:6 acyl chains originate from an $\omega 3$ fatty acid, docosahexaenoic acid (DHA).

Ratio images of LPE 20:4 and LPE 22:4 to LPE 22:6 indicate higher relative abundance of $\omega 6$ to $\omega 3$ LPEs in the striatum ventral region compared to the outer cortex (*Figure 2D–F*), consistent with the ratio distribution of arachidonic acid to docosahexaenoic acid (*Figure 2G*). Compared to diffuse non-ratio images of individual LPEs, the ratio images of $\omega 6$ to $\omega 3$ LPEs show an enhanced brain structure with markedly sharper definition. Despite variable LPE values imaged between COX10 KO and WT brains, overall LPE ratios were unchanged between genotypes. Note that we purposely chose this set of non-ideal section quality and imaging data to demonstrate the clear benefit of ratio imaging in obviating potential sample processing artefacts, thus improving image quality for data interpretation.

## Targeted metabolite ratio imaging may serve as a proxy for spatial enzyme and pathway activities among similar regions

For a given enzyme, pixel-wise specific enzyme activity may be inferred from pixel-by-pixel ratio imaging of substrate-product metabolite pairs. However, due to differences in ionization efficiency, comparison of pathway activity can only be limited to the equivalent pixel/regions of tissues from different biological groups, given the assumption that ionization efficiency is identical for equivalent

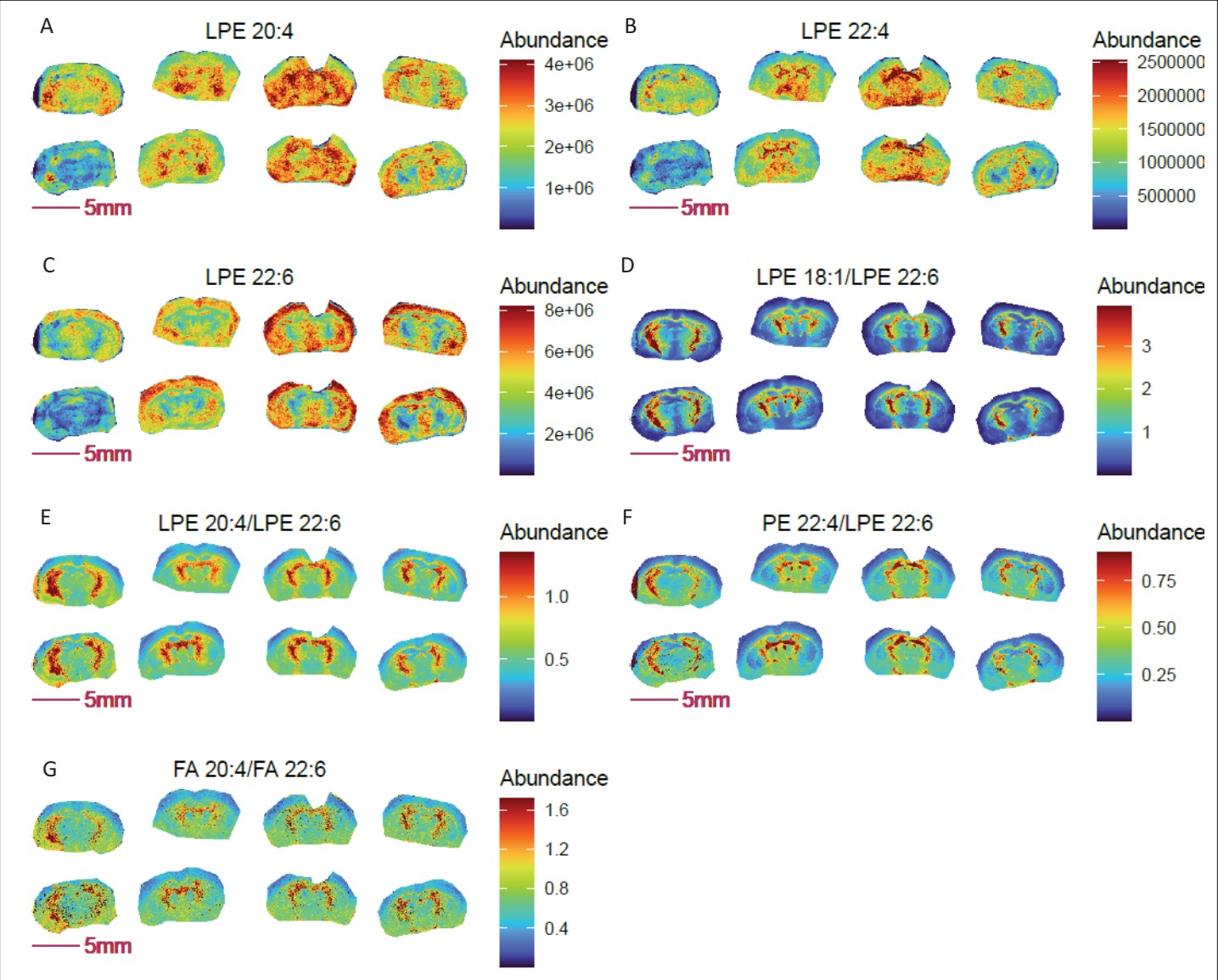

**Figure 2.** Pixel-by-pixel ratio image visualization reduces cryosection preparation artefacts and enables spatial metabolic discovery in COX 10 KO mouse brain (left 4 sections) compared to WT (right four sections). (**A-C**) Blurry and diffused images of LPEs with ω3 (**A–B**) and ω6 (**C**) polyunsaturated fatty acyl chains; (**D-F**) sharper ratio images of ω3 fatty acid containing LPE 20:4 and LPE 22:4 to ω6 fatty acid containing LPE 22:6; (**G**) ratio image of ω6 arachidonic acid (FA 20:4) to ω3 docosahexaenoic acid (FA 22:6) is consistent with the identified abundance ratio distributions for LPE 20:4 and LPE 22:4 to LPE 22:6.

The online version of this article includes the following source data for figure 2:

**Source data 1.** Source data contains pixel data for this figure.

pixel from different tissue sections (i.e. same cell type and microenvironment). Notably, metabolites with similar functional structure in the same pathway are better fit for this application. For instance, fatty acids with different chain length, phospholipid with same head groups, nucleotide phosphates with different phosphorylation status are expected to have similar ionization efficiency in the same tissue pixel/region. Examples of metabolite ratios that may serve as readouts for comparative enzyme activities include glutamine to glutamate for glutaminase (GLS), aspartate to asparagine for asparagine synthase (ASNS), aspartate to N-acetylaspartate for N-acetyltransferase 8 (NAT8L), inosine monophosphate (IMP) to adenosine monophosphate (AMP) for AMP deaminase (AMPD), lactate to pyruvate for lactate dehydrogenase (LDH), N-acetyl-aspartyl-glutamate (NAAG) to N-acetyl-aspartate for NAAG synthetase (NAAGS), carnosine to histidine for carnosine synthase (CARNS), hexose to

hexose phosphate for hexose kinase (HK), glutamate to oxyproline for 5-oxoprolinase (OPLAH), and glutathione disulfide to glutathione for glutathione peroxidase (GPX). Note that ratio image serves as a hypothesis generation tool, an orthogonal tool may be needed for a higher degree of biological confidence.

For lipid-related metabolites, ratios between different numbers of fatty acid double bonds for a given chain length (C16, C18, C20) would imply fatty acid desaturase (SCD) activity. Similarly, the fatty acid synthase (FAS) and PLA activities are reflected in the ratios of fatty acids with 2 carbon differences (C16 to C18, for example) and those of phospholipids to their corresponding lysophospholipids. Fatty acids are synthesized through metabolic pathways that include desaturation and elongation, sequentially producing a variety of long-chain saturated, monounsaturated, and polyunsaturated fatty acids (SFA, MUFA, and PUFA, respectively). Palmitic acid (FA 16:0) and stearic acid (FA 18:0) are the most common and abundant long chain saturated FAs in food and the human body. MUFAs synthesized from SFAs (via SCDs) are key components of phospholipids, triglycerides, and cholesterol esters that modulate cell membrane fluidity.

Obviously, pixel-by-pixel metabolite ratios are only possible for metabolite pairs detected and imaged within the same pixel of a given cryosection. MALDI-MSI using NEDC matrix and negative ion mode can detect all of the above-mentioned product-substrate pairs. Washing with acidic methanol significantly enhances the detection of phosphate-containing metabolites involved in energy metabolism (*Lu et al., 2023*). Alternatively, using 9-Aminoacridine (9AA) as a matrix, purine and pyrimidine nucleotide mono-, di- and triphosphates can be imaged to obtain pixel-by-pixel ratios of nucleotides with different phosphate levels for assessment of relative phosphorylation status within a tissue.

*Figure 3* demonstrates the benefit of imaging metabolically relevant metabolite ratio pairs in assessing regional gradients in enzymatic and metabolic pathways, as exemplified in E8.5 embryos at 10 µm (cellular) spatial resolution. Consistent with a role for the visceral yolk sac in providing nutrients, including glucose for embryo development, enhanced glycolytic activity was seen in the amnion and visceral yolk sac (*Figure 3A-B*). Furthermore, higher abundances of the desaturase product FA 18:1 relative to FA 18:0 (reflecting SCD activity) and of phophatidylethanolamine (PE) 38:4 relative to LPE 18:0 (reflecting PLA2 activity) are predominantly located in the allantois, head mesenchyme, and especially in the hindbrain neuroepithelium for FA 18:1 to FA 18:0 (*Figure 3C-H*). In contrast, the ratios of FA 16:0 relative to its elongation product FA 18:0, and DHA ($\omega$3) relative to AA ($\omega$6) fatty acids, showed opposite spatial distribution patterns (*Figure 3D-E*). Of note, similar $\omega$3 to $\omega$6 fatty acyl chain composition was also observed for LPEs (*Figure 3F-G*). Interestingly, PLA1 and PLA2 appear to have quite different compartmented activities among yolk sac, amnion, and head mesenchyme (*Figure 3G-H*). These data offer promising and useful information for study of compartmented metabolism during embryo development at the single cell level, demonstrating the power of metabolite ratio imaging in spatial assessment of metabolic activity.

As an additional example, we present mouse hippocampus MSI findings, which show that ratio imaging may can provide additional cell-type specific metabolic information at cellular (10 µm) spatial resolution compared to individual metabolite imaging alone. For example, to assess the ATP phosphorylation status. Notably, neurons have higher energy requirements and prefer mitochondrial oxidative phosphorylation as their dominant energy source. In accord with this expectation, higher ATP/ADP, ADP/AMP, IDP/IMP ratios (*Figure 3J–L*) are observed in neurons of mouse hippocampus, while indicators of oxidative stress (*Figure 3I*) and fatty acid synthesis (*Figure 3P*) were unchanged in neurons compared to other cell types. Elevated SCD and AMPD activities were also predominantly observed in neurons (*Figure 3M–N*) compared to other cell types, consistent with SCD being a novel regulator of neuronal cell proliferation and differentiation and their reported immunocytochemical results (*Knecht et al., 2001*). Although a higher ratio of $\omega$3 to $\omega$6 fatty acids predicts improved hippocampus-dependent spatial memory and cognitive status in older adults (*Andruchow et al., 2017*), ratio imaging surprisingly shows that these neurons display a relatively lower level of DHA ($\omega$3) to AA ($\omega$6) fatty acids vs. other cell types in mouse hippocampus. Thus, ratio imaging findings have the potential to challenge existing concepts and guide further investigation.

Adipose tissues store body fat as neutral triglyceride and represent the chief energy reservoir in mammals. We previously reported that COX10 KO mice are typified by decreased white adipose stores, accelerated lipolysis with increased free fatty acid deposition and low leptin levels with decreased food intake (*Southwell et al., 2023*). We have since applied lipid and fatty acid ratio

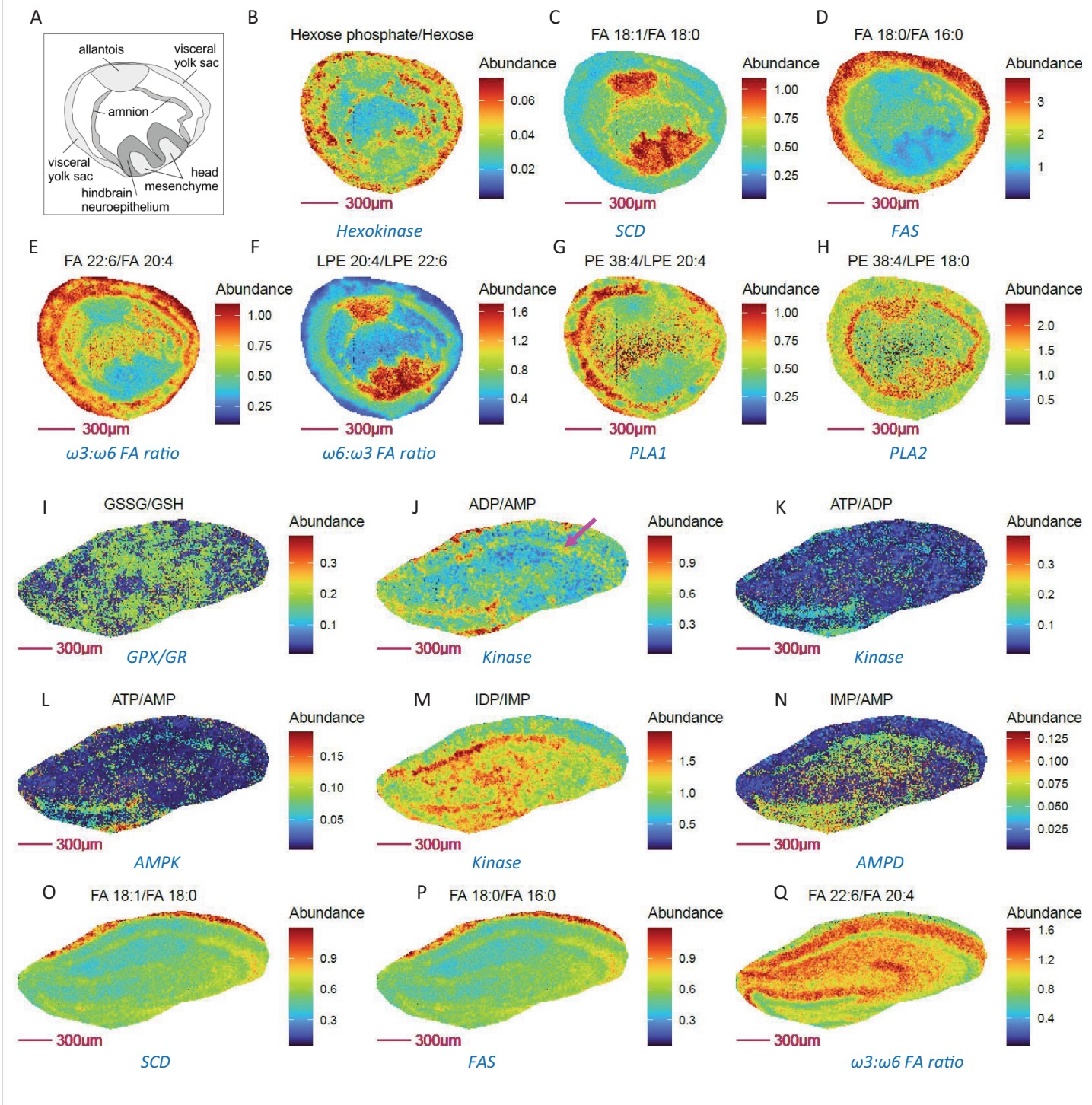

**Figure 3.** Metabolically relevant metabolite ratio pairs may probe enzymatic and metabolic pathway activity in E8.5 mouse embryo cryosections and mouse hippocampus at 10 μm spatial resolution. (**A**) Annotated drawing of E 8.5 embryo tissue, derived from the section adjacent to imaged section. (**B-H**) Ratio of substrate/ product pairs of glycolysis, fatty acid and lipid metabolism that imply compartmented activities for hexokinase (**B**), fatty acid desaturation (**C**), fatty acid chain elongation (**D**), fatty acid composition (**E–F**) and phospholipase activity (**G–H**) in E 8.5 embryos. (**I-Q**) Ratio images showing relative oxidative stress (**I**), adenine nucleotide energy levels (**J–L**), purine nucleotide cycle activity (**M–N**), fatty acid desaturation (**O**), synthesis (**P**), $\omega 3 : \omega 6$ FA ratio (**Q**) in neurons (purple arrow), compared to other cells in mouse hippocampus.

The online version of this article includes the following source data for figure 3:

**Source data 1.** Source data contains pixel data for this figure.

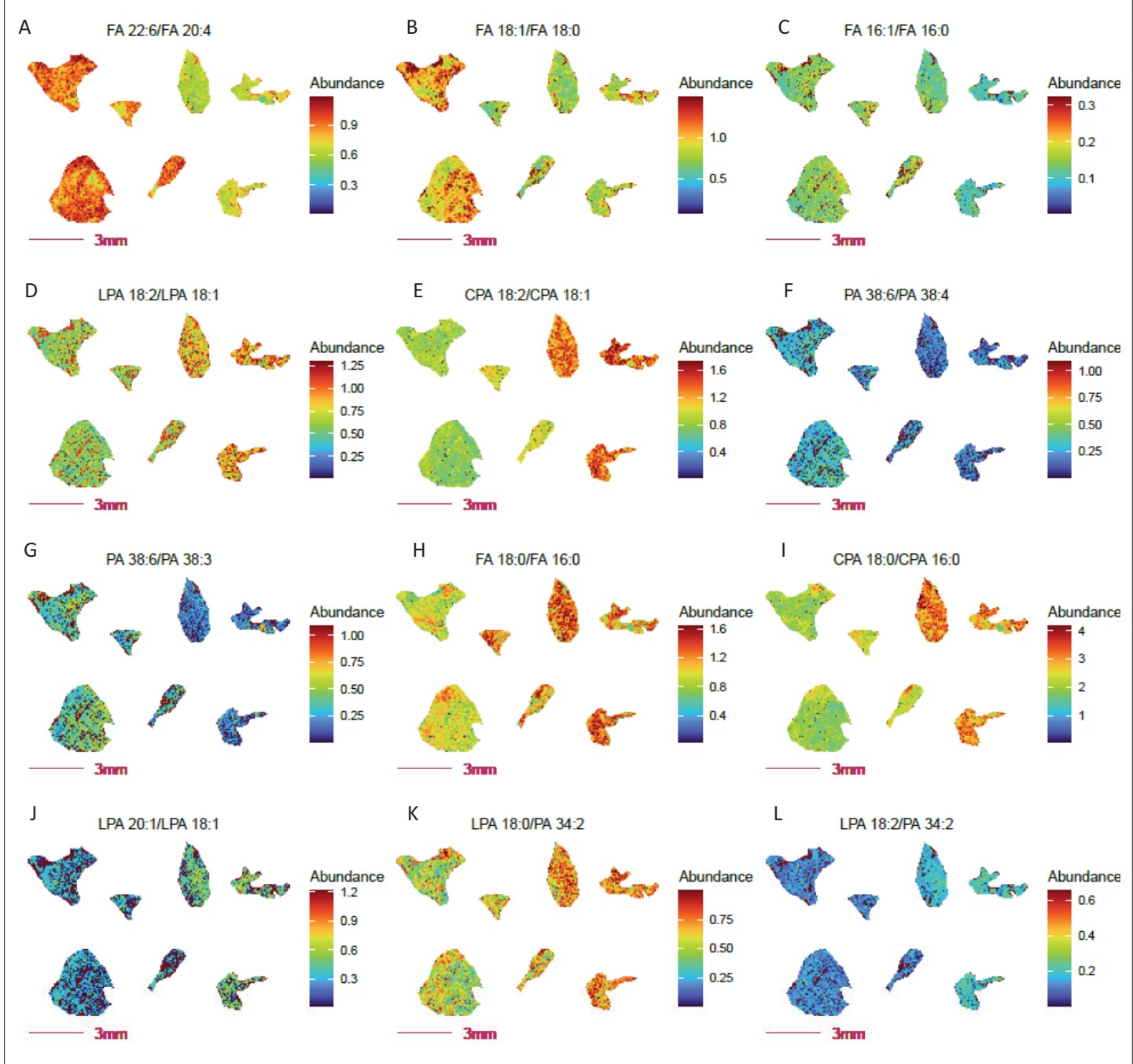

**Figure 4.** Altered fatty acid and lipid metabolism probed by ratio imaging in adipose tissue from COX 10 KO vs.WT mice. COX 10 KO, left 4 sections; WT, right 3 sections. Imaging data was acquired at 80 µm spatial resolution. (**A**) Decreased abundance ratio of FA 22:6 (DHA, ω3 fatty acid) to FA 20:4 (AA, ω6 fatty acid) in COX 10 KO adipose tissue. Panels (**B-C**) Decreased fatty acid FA 16:0 and FA 18:0 desaturation in COX 10 KO adipose. (**D-G**) Altered desaturated fatty acid abundance ratios for lysophospolipids and phospholipids in COX 10 KO adipose tissue. Panels (**H-J**) Elevated fatty acid synthesis inferred by free fatty acid and phospholipids ratios, considering FA 18:0 to FA 16:0 (**H–I**) and FA 20:1 to 18:1 (**L**) in COX 10 KO adipose. Panels (**K-L**) Elevated PLA2 (**K**) and PLA1(**L**) activities implied by the elevated ratio of Lyso PA to PA in COX 10 KO adipose.

The online version of this article includes the following source data for figure 4:

**Source data 1.** Source data contains pixel data for this figure.

imaging to evaluate potential metabolic and enzymatic activities that may be responsible for the observed differences between COX10 KO and WT adipose tissue (*Figure 4*). As expected, distinct fatty acid and lipid profiles point to altered $\omega 3$ (DHA) to $\omega 6$ (AA) fatty acid composition (*Figure 4A*), fatty acid and lipid desaturation (*Figure 4B–G*), fatty acid synthesis (*Figure 4H–J*) and phospholipid hydrolysis (*Figure 4K–L*) in KO vs. WT adipose. Lower adipose $\omega 3$ to $\omega 6$ ratios have been associated with inflammation (*Bakker et al., 2023*), obesity (*Pinel et al., 2021*), altered gut microbiota (*Portela et al., 2023*; *Wall et al., 2009*) and other health problems (*Chiusolo et al., 2023*; *Rix et al., 2022*). These ratio imaging findings led us to speculate that the diminished $\omega 3$ to $\omega 6$ ratio in COX10 KO mice arises from an altered gut microbiome, supported by unpublished preliminary findings. This emergent hypothesis from ratio imaging analysis raises the clinically important possibility that supplementation of human COX10 mutant patients with $\omega 3$ fatty acids may effectively modify the microbiome to restore normalized adipose tissue composition and food intake. Note that ionization efficiencies and cryosection collecting method can impact the relative abundance of metabolites used in ratio computation and thereby the numeric ratio value itself. For example, the ATP/AMP ratio can change drastically from tissue collection, so can lysophospholipid to phospholipid ratio. Taken together, these results demonstrate the utility of ratio imaging as a valuable add-on MSI approach to probe for potential regiospecific changes in cellular metabolic activities.

## Metabolite ratios uncover genotype-specific and spatially resolved tissue regions

Conventional MSI depicts relative spatial abundance of metabolite features in tissue regions. For defined structures in tissues such as brain, kidney and prostate, metabolite images can potentially recognize discrete tissue structures, under the condition that ionization efficiency and/or absolute abundance differences do indeed exist among these structurally defined loci. In contrast, pixel-by-pixel metabolite ratio imaging almost invariably resolves distinct tissue structures but also uncovers fine structure and anatomically unrecognized regions not revealed by individual metabolite imaging alone.

Using COX10 KO and WT mouse coronal brain sections as an example, we observed that individual glutamate, aspartate and glutamine distribution images failed to show a clear structural distribution, (*Figure 5A–C*). However, ratio imaging revealed a 1.59-fold increase in aspartate to glutamate ratio in an unusual 'moon arc' region across the amygdala and hypothalamus (mean abundance 0.563 in 6345 pixels) relative to the rest of the coronal brain (mean abundance 0.353 in 45742 pixels, *Figure 5D*). Similar but different arc-like structures are encompassed within the ventral thalamus and hypothalamus, wherein glutamate to glutamine ratio show a 1.63-fold increase in intensity compared to the rest of the brain (mean abundance of 0.695 in 7108 pixels vs 0.428 in 44979 pixels, *Figure 5E*), while relative glutamine to aspartate abundance appears enriched in the striatum (*Figure 5F*). Thus, ratio imaging not only improves image quality for regional/cell-selective assessment of metabolite distributions but also offers an opportunity to discover new differentially metabolic regions with biological relevance. Note that the enrichments of aspartate to glutamate and glutamine to glutamate near the thalamus, nucleus accumbens (NAc) and hypothalamus (and visualization of their reciprocal ratio images) were robust and consistently reproduced in MALDI-MSI experiments using an independent cohort of COX10 KO and WT mouse brains (*Figure 5G–H*). Enrichment of glutamate to aspartate in the cortex region was further confirmed in horizontal and sagittal brain sections from random mouse brains (*Figure 5I–J*).

Considering the above, it is notable that aberrations in glutamate, glutamine, and aspartate homeostasis are all implicated in neurological disease (*Hertz and Rothman, 2017*; *Andersen et al., 2022*). Aspartate serves as an amino donor for glial glutamate formation (*Pardo et al., 2011*), and the hippocampal aspartate to glutamate ratio has been related to changes in brain glucose (*Szerb and O'Regan, 1985*). Glutamine is synthesized from glutamate and ammonia by the catalytic action of glutamine synthetase (GS) in glial cells of the brain, while it is also taken into neurons and generates glutamate and ammonia by the catalytic action of glutaminase. The glutamate-glutamine cycle is closely linked to cellular energy metabolism (*Andersen et al., 2022*), and ratios between neurotransmitters (GABA, glutamine, and glutamate) have been used in MRS studies to probe neurological disease mechanisms and progression (*Kantorová et al., 2022*; *Kantorová et al., 2021*). Specifically, ratios among glutamine, glutamate, and aspartate have been implicated in memory function (*Limón*

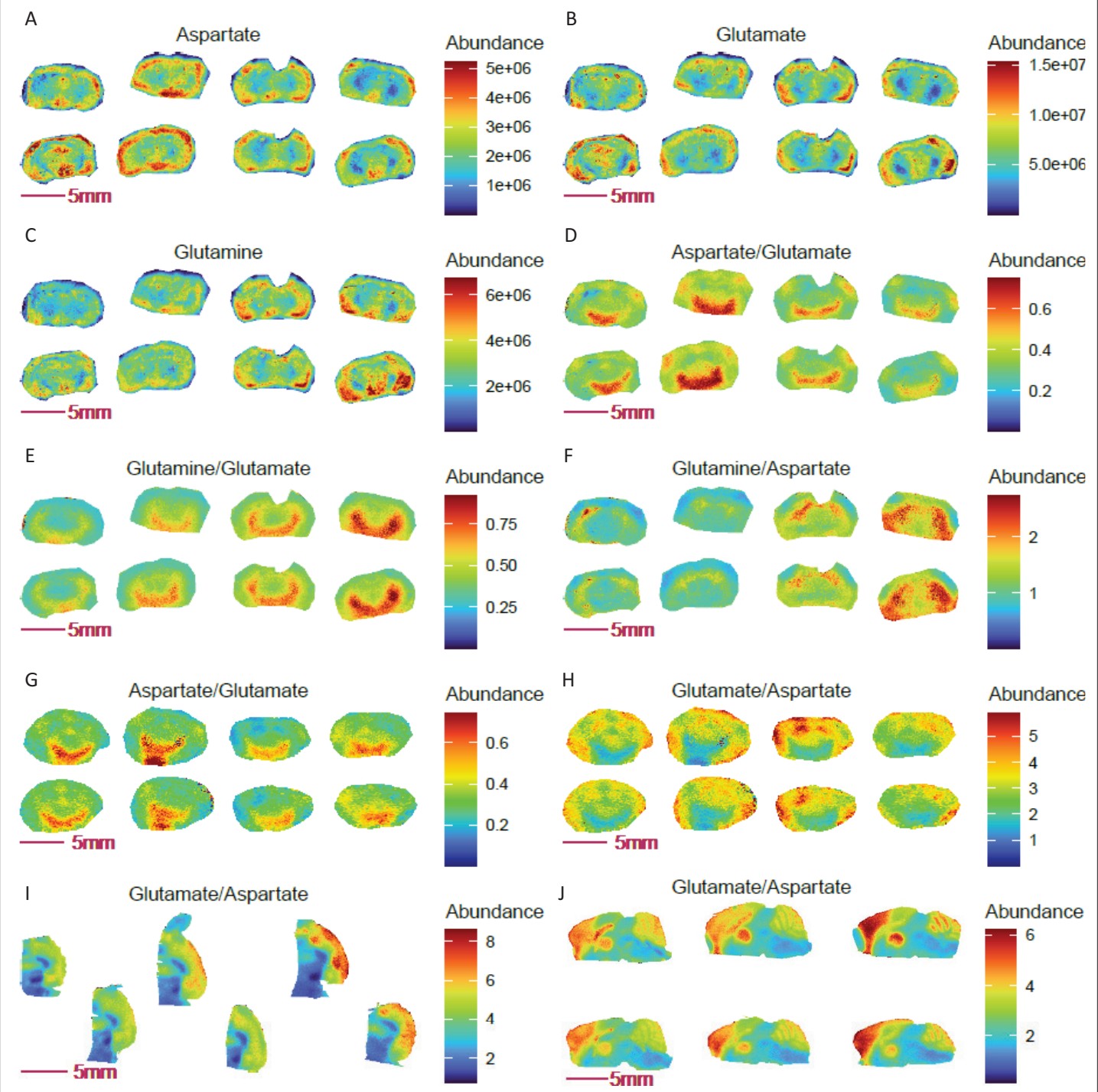

**Figure 5.** Ratio imaging uncovers novel mouse brain regions with genotype-specific and spatially resolved ratios of aspartate, glutamate, and glutamine. (**A-C**) Images of glutamine, glutamate and aspartate in coronal sections of COX 10 KO (left 4 sections) and WT (right 4 sections) mouse brains. (**D-F**) Differential abundance ratios of aspartate, glutamate and glutamine observed in defined regions of the COX 10 KO and WT brains (p<0.0001, two-tailed Student t-test with Benjamini-Hochberg correction). Panels **G-H**: Reciprocal and differential aspartate to glutamate ratio and Arc-like NAC containing region was repeated in a separate group of COX 10 KO (left 4 sections) and WT (right four sections) mouse brains. (**I-J**) Enrichment in the glutamate to aspartate ratio in the cortex region of mouse brain, as revealed in horizonal and sagittal sections.

The online version of this article includes the following source data and figure supplement(s) for figure 5:

**Source data 1.** Source data contains pixel data for this figure.

*Figure 5 continued on next page*

*Figure 5 continued*

**Figure supplement 1.** Ratio of glutamate to glutamine and Fe$^{2+}$ abundance are co-enriched in the outer cortex of the COX10 KO (right 4 sections) and WT (left 4 sections) brain.

**Figure supplement 1—source data 1.** Source data contains pixel data for this figure.

*et al., 2021*), schizophrenia, depression, alcoholism and motivated performance via NAc (*Hangel et al., 2020*). Given the biological significance of these ratios, spatial ratio imaging can reveal previously unrecognized ROI and relative abundance ratio differences between disease and normal physiological states. In COX10 KO and WT brains, differential aspartate to glutamate and glutamine to glutamate ratios point to an interesting linkage between muscle health and brain neurotransmitter signaling: mitochondrial myopathy alters glutamine, glutamate and aspartate signaling near the NAc. This unusual finding is in accord with and supportive of our recent discovery of a coordinated multiorgan metabolic response that extends to the brain and is anticipated to play a role in human mitochondrial myopathy disease states (*Southwell et al., 2023*).

## Spatial correlation of ratio and non-ratio pixel-by-pixel data as an additional discovery tool

Combining pixel-by-pixel ratio and non-ratio data offers an additional tool for spatial metabolism discovery. Despite that MALDI MSI is a soft ionization technique, it generates various adduct ion clusters and possible fragment ions from common neutral losses. Unknown non-ratio metabolite feature data in a given ROI can be combined with known metabolite ratios, pixel-by-pixel, to discover correlations between unknowns and metabolite ratios that inform on a specific enzyme activity or metabolic pathway. This application can reveal unrecognized but significantly correlated pathway metabolites

**Table 1.** Spearman correlation of an unknown with ratio and non-ratio entities.

| Entity | Unknown [M-H] 375.23059 |
|---|---|
| CPA 18:1 | 0 86090633 |
| CPA 16:0 | 0 85862853 |
| Docosahexaenoic Acid | 0 84761078 |
| Palmitic Acid | 0 80091519 |
| LPA 18:1 | 0 78455912 |
| LPA 16:0 | 0 77663768 |
| Oleic Acid | 0 77371792 |
| PA 34:2 | 0 73231961 |
| PA 36:3 | 0 72919804 |
| Palmitoleic Acid | 0 72724047 |
| CPA 18:0 | 0 70128825 |
| LPE 16:0 | 0 69156441 |
| PA 38:6 | 0 68799022 |
| LPA 18:0 | 0 67264794 |
| 558.4283 | 0 67163649 |
| PE P 16:0 | 0 66456185 |
| Linoleic Acid | 0 65773742 |
| PI 36:2 | 0 65460109 |
| LPA 18:2/LPA 18:1 | –0 53780801 |
| PA 36:1/LPA 18:1 | –0 4549225 |
| PA 36:4/LPA 16:0 | –0 42353603 |

**Table 2.** Brain glutamate to glutamine ratio positively correlates $FeCl_2$ using combined ratio and non-ratio pixel data and spearman correlation.

| Metabolite or Metabolite ratio | Glutamate/Glutamine |
| --- | --- |
| Glutamate/Glutamine | 1 |
| N-Acetylaspartate/Glutamine | 0.69140746 |
| Glutamate/Glucose | 0.68318508 |
| Aspartate/Glutamine | 0.617246741 |
| Aspartate/Glucose | 0.571634522 |
| N-Acetylaspartate/Glucose | 0.487447557 |
| Glutathione/Glutamine | 0.482704981 |
| Glutamate/Malate | 0.477826203 |
| Glutamate/NAAG | 0.418208761 |
| Glutathione/Glucose | 0.414914055 |
| Glutamate/Taurine | 0.408270132 |
| Aspartate/NAAG | 0.40560898 |
| FeCl2 | 0.40137113 |

and metabolite ratios, thus enabling recognition of unknown metabolites with relevance to specific pathways of interest.

Using COX10 KO and WT mouse adipose as an example, the unknown [M-H] 375.23059 was negatively correlated with the ratios of PA 36:4 to LPA 16:0 and PA 36:1 to PA 18:1, indicating that the abundance of this unknown metabolite may be positively associated with PLA1 and/or PLA2 activity (*Table 1*). Given that PLA1 and PLA2 generate different products, with PLA1 dominantly yielding unsaturated lysophospholipids and PLA2 yielding saturated lysophospholipids, an observed negative correlation of the unknown with LPA 18:1 to LPA 18:0 ratios hints that it may originate as a product of PLA2. Indeed, direct correlation of this unknown with lipid metabolites suggests it could be a reduction product of CPA 16:0 after loss of one oxygen atom, matching the formula $C_{19}H_{37}O_5P$ with 0.013 ppm mass accuracy. While it remains to be established that this unknown is a bona fide reduction product of CPA 16:0, the example illustrates how pixel-by-pixel correlations of unknown metabolite masses with metabolite ratios can aid potential discovery of pathway-relevant metabolites.

Another example of correlation-based discovery comes from consideration of COX10 KO and WT mouse brain sections, where calculating the Spearman correlation of combined pixel data from both individual metabolites and metabolite ratios revealed an unexpected positive association of $FeCl_2$ with the glutamate to glutamine ratio, along with other related metabolite ratios that include aspartate, NAA, glucose and glutathione (*Table 2*, *Figure 5—figure supplement 1*). Conversely, glutamine to glutamate ratios showed significant negative correlation with $FeCl_2$ and the above-mentioned metabolites (*Supplementary file 2*), suggesting these correlations are neither random nor an artefact of missing value annotation. Notably, free iron is recognized as a key mediator of glutamate excitotoxicity in spinal cord motor neurons (*Yu et al., 2009*). Significant associations between peripheral markers of iron metabolism and glutamate with glutamine in relation to total creatine (Glx:tCr) concentration have been reported in female human brains (*Burger et al., 2020*). Glutamate facilitates divalent metal transporter 1 (DMT1) flux and subsequent increases in brain free iron content (*Yu et al., 2015*). A positive relation between iron and glutamate in the brain and other organs has also been reported (*Pardo et al., 2011*; *Szerb and O'Regan, 1985*; *Yu et al., 2009*; *Burger et al., 2020*; *McGahan et al., 2005*; *Mittal et al., 2003*).

Note that this correlation with $FeCl_2$ was observed in all brain sections regardless of genotype. Ferrous iron, measured as its chloride adduct when detected with NEDC matrix, likely arises from both free labile iron and protein-bound iron. Decreased $FeCl_2$ levels in combination with decreased glutamate in the cortex region of COX10 KO vs. WT brain is an unexpected but intriguing finding. While aging is associated with increased brain iron through cortex-derived expression of the iron regulatory

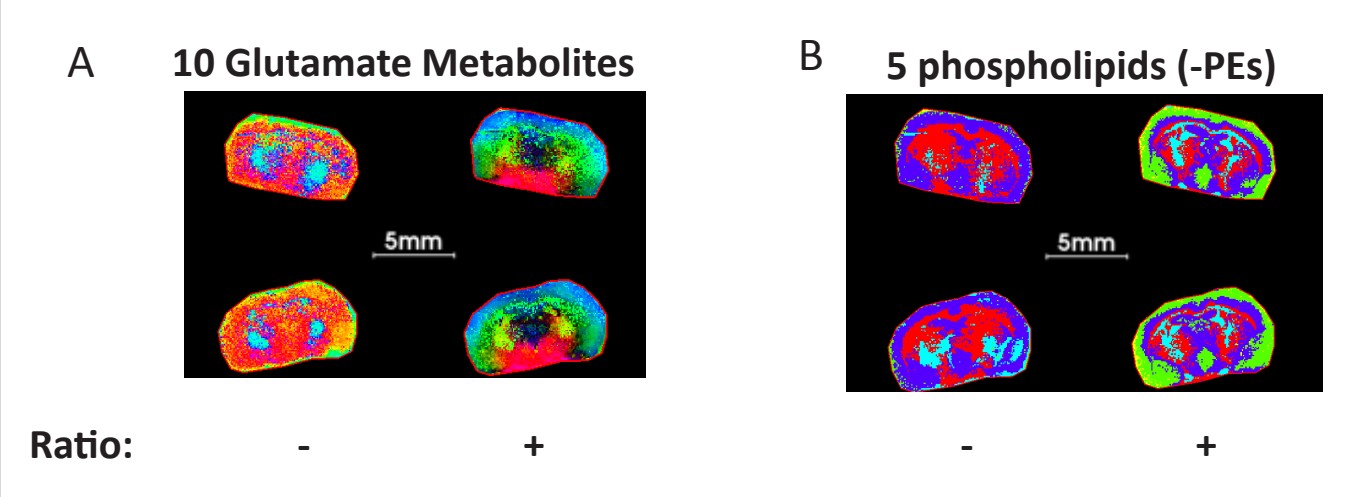

**Figure 6.** PCA segmentation of mouse brains comparing ratio and non-ratio metabolite pixel data. (**A**) PCA considering the top 5 components from ratios of 10-glutamate related metabolites that reveal an unanticipated arc-like regional distribution within the hypothalamus, striatum and NAc, that is not detected by consideration of only non-ratio metabolite data. (**B**) PCA considering the top 5 components 5-LPE metabolites showing fine brain structure that is not fully represented by only non-ratio metabolite segmentation.

hormone hepcidin (*Sato et al., 2022*), calorie restriction down-regulates hepcidin expression (*Wei et al., 2014*). In accord with decreased $Fe^{2+}$ and glutamate to glutamine ratio, COX10 KO mice exhibited decreased food intake and adopted a metabolic state resembling caloric restriction (*Southwell et al., 2023*). Collectively, pan-correlation of targeted metabolite ratios with non-ratio data serve as an additional imaging tool for data interpretation and hypothesis generation.

## Metabolite ratios for PCA tissue segmentation in complex tissue sections

Spatial segmentation is a popular application in MSI to cluster regions with similar metabolite abundance profiles. Unsupervised MSI segmentation methods such as PCA, K-means, bisecting K-means and HCA are all often used to delineate underlying tissue structures from high-dimensional MSI data without prior knowledge of sample anatomical information. Most segmentation approaches rely on unsupervised clustering algorithms that can readily result in the generation of biologically unrealistic tissue structures. Various algorithms for integrating partial or prior structural information with unsupervised clustering have been reported to improve segmentation results (*Guo et al., 2022a*; *Shi et al., 2022*; *Guo et al., 2022b*; *Zou et al., 2023*; *Baars et al., 2021*; *Guo et al., 2021*; *Hu et al., 2021*).

To date, tissue segmentation utilizing pixel-by-pixel metabolite ratio data for imaging differential metabolic zones has not been reported. We hypothesized that metabolite/feature ratios may generate more meaningful biologically relevant ROIs due in part to reduction of systematic experimental noise. Mouse brain sections serve as a good example for the utility of this approach. As expected, consideration of 10 different glutamate/glutamine-related metabolites for PCA analysis (*Supplementary file 3*) with ratios involving the top 5 components generated an interesting arc-shaped region that encompasses the hypothalamus, striatum and NAc, while non-ratio metabolite data did not (*Figure 6A*). This metabolite ratio PCA result was used to provide an RGB image capturing the structure of the dataset by mapping the first of the five principal components. Apparently, this segmentation result captures and maintains the enriched arc-like regions shown previously in neurotransmitter ratio images (*Figure 5D–E and G*). Since glutamate plays key roles linking carbohydrate and amino acid metabolism via the tricarboxylic acid (TCA) cycle, as well as in nitrogen trafficking and ammonia homeostasis in the brain, this region likely exhibits unique glutamate-related metabolism and warrants further investigation. Likewise, a more defined outline of striatum and cerebellum in the coronal brain structure was obtained in PCA segmentation using the top 5 component metabolite ratios as well as 5 LPEs *Supplementary file 4* but was not observed with these 5 LPEs alone (*Figure 6B*).

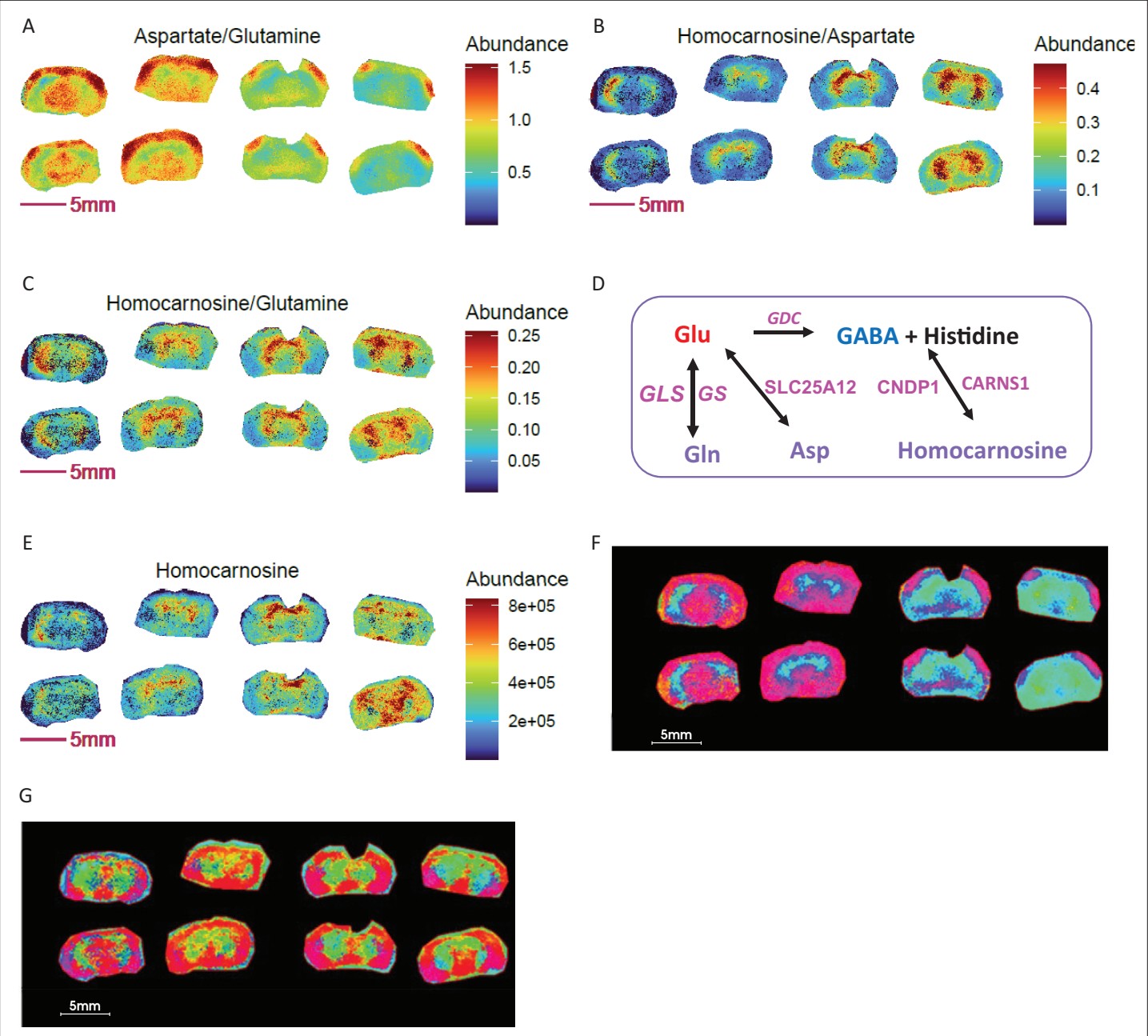

**Figure 7.** Metabolite ratios for segmentation and biomarker discovery in segmented ROIs. KO: right 4 sections; WT: left 4 sections. (**A-C**) Ratio images of homocarnosine, glutamine and aspartate. (**D**) Enzymatic regulation and interconversions among brain glutamine, aspartate and homocarnosine. (**E**) Image of homocarnosine. (**F**) PCA with top 5 components and using ratios of homocarnosine, glutamine and aspartate shows differential ROI abundance shown in cyan blue in larger area of striatum, thalamus hypothalamus regions in KO compared to WT. (**G**) PCA with top 5 components using homocarnosine, glutamine, and aspartate pixel data show indistinguishable ROIs between genotypes.

The online version of this article includes the following source data for figure 7:

**Source data 1.** Source data contains pixel data for this figure.

As metabolite ratio data can segment tissue into novel ROIs, we considered the possibility that differential metabolite ratios can serve as markers to distinguish genotypes, either independently or when combined with individual metabolites. In COX10 KO and WT mouse brains, ratios among three neurotransmitter-related non-glutamate metabolites (glutamine, aspartate, and homocarnosine) show clear differences within the cerebral cortex for glutamine to aspartate ratios (*Figure 7A*) and

within the ventral striatum for homocarnosine to glutamine and homocarnosine to aspartate ratios (*Figure 7B–C*).

The interrelationship of glutamine, aspartate and homocarnosine is shown in *Figure 7D*. Homocarnosine is a naturally occurring imidazole-containing dipeptide uniquely present in the brain, likely in a subclass of GABAergic neurons (*Rothman et al., 1997*). It is synthetized from γ-aminobutyric acid (GABA) and histidine via homocarnosine synthase (CARNS1) and can be hydrolyzed to histidine and GABA by carnosine dipeptidase 1 (CNDP1), which is exclusively expressed in brain (*Figure 7D*). Homocarnosine is recognized to play a role in mediating neuronal activity during satiation and starvation (*Filipović et al., 1993*). It is decreased in the brain during cold adaptation (*Bondarenko et al., 1985*) and increased during alcohol intoxication (*Kukharenko et al., 1990*). Additionally, aspartate and glutamate can be exchanged across the inner mitochondrial membrane by mitochondrial carrier protein SLC25A12 (Aralar1), mainly distributed in brain and skeletal muscle for the transport of aspartate from mitochondria to cytosol, and in the transfer of cytosolic reducing equivalents into mitochondria as a member of the malate-aspartate shuttle (*Jalil et al., 2005*). Given the interrelationship of these three metabolites and their important roles in maintaining brain neurotransmitter pools, their ratios could report on neurotransmitter homeostasis. Indeed, it has been shown that brain function can be estimated from the ratio of glutamine to homocarnosine in cerebrospinal fluid (*Ohtsuka and Takahashi, 1983*). An early report showed that homocarnosine is elevated in monkey brain in the setting of a protein-deficient diet, whereas aspartate and glutamate levels are unchanged (*Enwonwu and Worthington, 1973*). Similarly, in COX10 KO and WT brains, we observed no difference in individual aspartate or glutamate levels (*Figure 5A–B*), while striatal homocarnosine (*Figure 7E*) and glutamine (*Figure 5C*) were elevated. As COX10 KO mice displayed altered food intake, muscle mass and elevated starvation response (*Southwell et al., 2023*), an elevated homocarnosine to glutamine and glutamine to glutamate ratio (*Figure 5F*) could indicate increased synthesis of homocarnosine via the glutamine-glutamate-GABA axis. These results indicate that ratios among the three metabolites could serve as alternative markers to distinguish these murine genotypes.

To provide additional spatial detail, we performed PCA segmentation on COX10 KO and WT brains using ratio and non-ratio pixel data for homocarnosine, glutamine and aspartate with the goal of learning whether differential ROIs exist for these genotypes. In this PCA segmentation, the overview RGB image maps the first three principal components to the red, green and blue channels respectively. Comparing PCA overview images from ratio (*Figure 7F*) to non-ratio (*Figure 7G*), PCA results from ratios showed cyan blue located in larger areas of striatum and thalamus regions in KO relative to WT brains, while non-ratio PCA did not. These results reveal distinguishable differences in abundance in the cyan region of the KO and WT brain regarding interconversion and homeostasis of the three neurotransmitter-relevant metabolites. Therefore, metabolite ratios could serve as better candidates for ROI-based disease metabotyping.

## UMAP using metabolite ratios enables spatial metabotype mapping and recognizes novel ROIs

Unlike PCA analysis, which builds on a linear dimension reduction algorithm, uniform manifold approximation and projection (UMAP) is a relatively newer non-linear dimension reduction algorithm often used to visualize complex dataset while preserving local structure. UMAP projects dimensionally reduced data into 2D or 3D scatterplots. Samples close to each other in the scatterplot have similar fingerprints, while samples further away have progressively different profiles. It is often used in combination with other clustering algorithms such as HCA and K-mean clustering to identify subgroups in the projected space. UMAP has been widely used for omics data interpretation to find internal structure in mixed datasets, as well as for disease subtyping, single-cell biogy (*Rather and Chachoo, 2023*; *ElKarami et al., 2022*; *Do and Canzar, 2021*) and various molecular imaging applications.

Since metabolite ratio segmentation using PCA is less prone to section preparation artefacts in MALDI-MSI and enables genotype-specific tissue segmentation (*Figure 7F*), we sought to assess whether metabolite ratio UMAP and clustering could generate novel ROIs that enable new spatial metabotype mapping beyond that obtained from consideration of individual metabolites. UMAP analysis of ratios for 5 LPEs in COX10 KO and WT murine brain sections, followed by K-mean clustering, showed much more defined and biologically relevant ROIs than those derived from non-ratioed LPEs (*Figure 8*) that resemble PCA segmentation results (*Figure 6B*). This UMAP analysis used nine

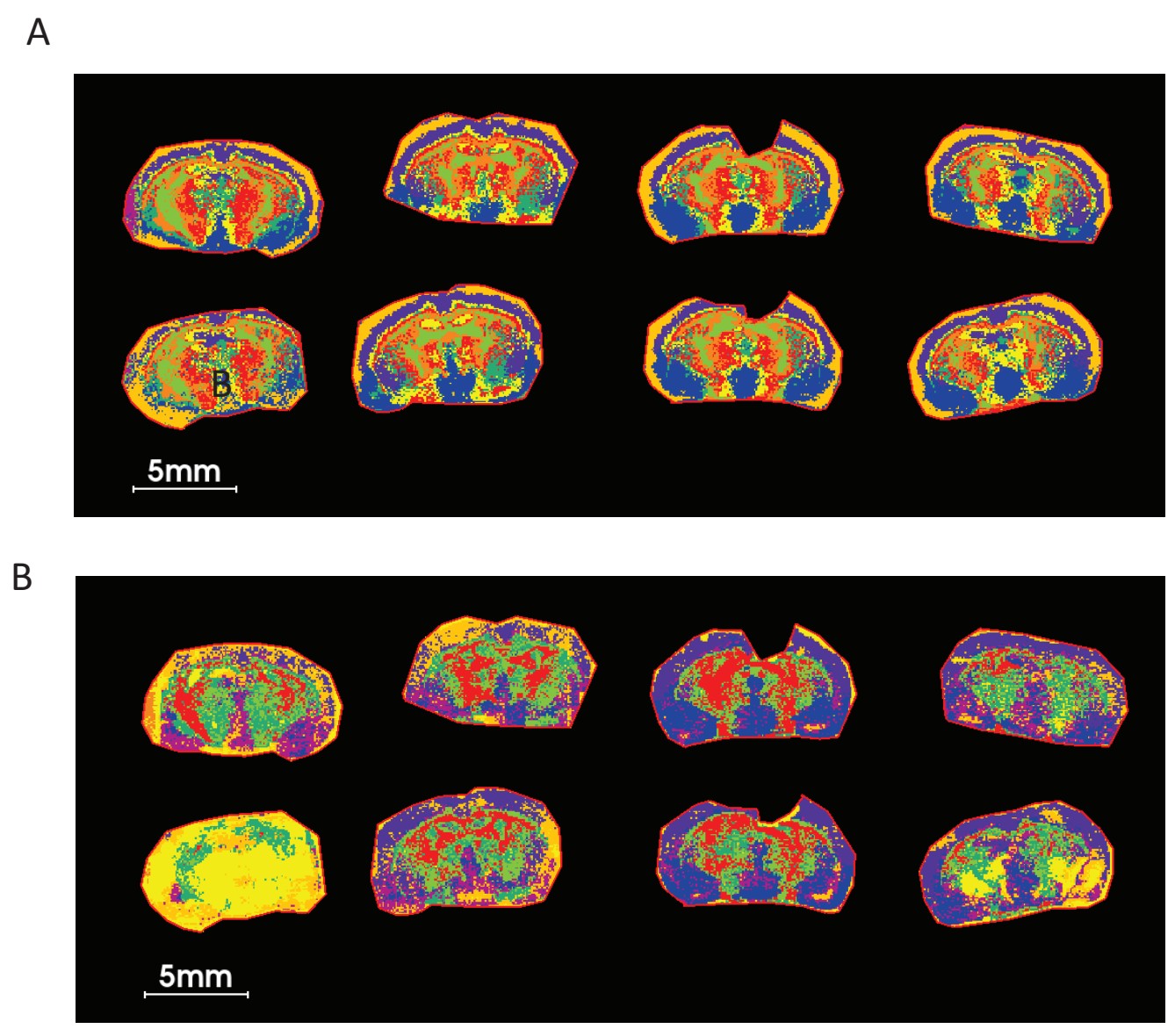

**Figure 8.** UMAP analysis using five LPEs ratio followed by K-mean clustering on COX KO and WT brain sections shows a much cleaner and biologically relevant ROIs compared to those of LPE non-ratio. (**A**) UMAP clustering using ratios of five LPEs. (**B**) UMAP clustering using five LPE pixel data. The list of five LPEs is shown **Supplementary file 4**. Nine neighbors, two components with Euclidean distance as metric measurement of similarities were used in this UMAP analysis.

The online version of this article includes the following source data and figure supplement(s) for figure 8:

**Figure supplement 1.** ROIs and UMAP scatter plots generated from non-ratio metabolite data.

**Figure supplement 1—source data 1.** Source data contains pixel data for UMAP score plot.

**Figure supplement 2.** ROIs and UMAP scatter plots generated from metabolite ratio data.

**Figure supplement 2—source data 1.** Source data contains data for UMAP score plot.

**Figure supplement 3.** Ratio images of NAA, NAAG and aspartate for UMAP-generated ROIs.

**Figure supplement 3—source data 1.** Source data contains ROI pixel data for this figure.

**Figure supplement 4.** Images of NAAG and neurotransmitter related metabolites visualized in UMAP-generated ROI 7.

**Figure supplement 4—source data 1.** Source data contains pixel data for this figure.

**Figure supplement 5.** Pixel-by-pixel correlation of glutamine to glutamate ratio with a compendium of all detected metabolites and ratios exclusively in UMAP-generated ROI 7, comparing WT (**A**) to COX 10 KO (**B**) brain.

*Figure 8 continued on next page*

*Figure 8 continued*

**Figure supplement 5—source data 1.** Source data contains correlation matrix data for this figure.

**Figure supplement 6.** UMAP using five LPE ratios generated ROIs consistent with brain anatomy in two separate experiments with horizonal and sagittal brain sections.

**Figure supplement 6—source data 1.** Source data contains data for UMAP score plot in this figure.

neighbors with two Euclidean components as metric measurements of similarity. ***Figure 8—figure supplements 1 and 2*** show the nine segmented ROIs from ratio and non-ratio UMAP and K-mean clustering analyses, along with scatterplots of the first two UMAP components (V1 vs. V2), projecting the distribution and similarity of each of nine segmented ROIs. While non-ratio segmented ROIs did not provide comparable and confident ROIs among replicates (***Figure 8—figure supplement 1***), the ratio-segmented data indicate a distinct LPE profile between the outer cortex (ROI 3), cortex (ROI 8), ventral thalamus (ROI 5) and striatal regions (ROI 2; ***Figure 8—figure supplement 2***). These ROIs generated by UMAP metabolite ratios enable visualization and a reproducible comparison of lipid metabolite and/or ratio abundance among these irregular and difficult-to-draw (i.e. with imprecise anatomical definition) regions. In addition, they provide unique ROIs for pixel-by-pixel metabolite correlation with other previously mentioned multivariate analyses (***Table 1***, ***Figure 1—figure supplement 1***).

Differences in metabolite abundance among ROIs can be difficult to visualize if the image color scale is set too wide. ROI images are better visualized by applying a relative image abundance scale set to a discrete ROI, rather than the entire tissue section. UMAP segmentation using ratios provides novel ROIs for such purpose. For example, murine brain imaging results showed decreased aspartate to NAA ratios visualizable in almost all 8 ROIs investigated with COX10 KO relative to WT, suggesting overall diminished expression or activity of NAT8L, the neuronal enzyme that produces NAA from acetyl-CoA and aspartate (***Figure 8—figure supplement 3A-H***). Moreover, NAAG, the most abundant dipeptide in the brain, can be synthesized from NAA and glutamate by NAAG synthase. NAAG to NAA and aspartate ratios were greater in ROI 7 of KO compared to WT brain (***Figure 8—figure supplement 3I-J***), suggesting elevated NAAG synthase activity in this defined region. Of note, NAT8L mRNA oxidation is linked to neurodegeneration in multiple sclerosis (***Kharel et al., 2023***) and overexpression-induced vulnerability to depressive behavior in mice (***Uno et al., 2019***). NAAG functions as a retrograde neurotransmitter and is released in response to glutamate, providing post-synaptic neurons with a feedback mechanism to inhibit excessive glutamate signaling (***Morland and Nordengen, 2022***). Therefore, elevated NAT8L activity, indicated by an increased NAA to aspartate ratio, is consistent with observations that COX10 KO mice are less mobile and relatively inactive compared to WT. Reviewing the non-ratio images of these metabolites, NAAG is mainly enriched near ROI 7 visualized in whole section images (***Figure 8—figure supplement 4A***), consistent with higher abundance of NAAG, NAA and glutamine imaged exclusively in COX10 KO ROI 7 (***Figure 8—figure supplement 4B-D***). These changes contrast with the decreased aspartate and unchanged glutamate levels imaged in ROI 7 of KO brains (***Figure 8—figure supplement 4E-F***). Therefore, we infer that the COX10 KO brain exhibits increased NAA and NAAG syntheses as feedback to inhibit excessive glutamate signaling. In contrast, glutamate within the NAc showed no change compared to WT, potentially due to excess glutamate being used for synthesis of glutamine and NAAG, which are elevated in KO ROI 7 (***Figure 8—figure supplement 4C-D***). Consideration of these UMAP findings exemplifies the applicability of this strategy for MSI data interpretation.

UMAP derived ROIs can be applied to survey metabolite abundances, pixel-by-pixel, for correlation analysis in ROIs from a large combined dataset that includes metabolite ratio and non-ratio pixel data, as shown previously for whole tissue sections (***Tables 1 and 2***). When correlation is narrowed to a specific ROI and genotype, regional correlation results can be very different from global correlations. ***Figure 8—figure supplement 5*** shows pixel-by-pixel correlation of the glutamine to glutamate ratio, along with a compendium of all detected metabolites and ratios contained in a UMAP-generated ROI 7, comparing COX10 KO and WT brain. For improved correlation plot visualization, only significant correlations with Spearman correlation coefficients >0.4 are presented in ***Figure 8—figure supplement 5***. Glutamate to glutamine ratio positively correlated with glutamate, while negative correlations were observed for NAAG and NAAG/NAA in both WT and KO ROI 7 (***Supplementary file 5***).

These findings are consistent with existing knowledge that NAAG synthesis inhibits excess glutamate signaling.

Given that LPE ratios can be used as a tool for brain segmentation and to overcome variations in section quality, we asked if UMAP with the same 5 LPE ratios can similarly be applied for other types of brain section ROI segmentation. As predicted, UMAP generated ROIs are consistent with brain anatomy in two separate experiments depicted by horizonal and sagittal brain sections (*Figure 8— figure supplement 6*). Collectively, UMAP analyses using metabolite ratio imaging provide a novel and robust tool to spatially-map tissue microenvironments, revealing matched localized differences among test groups.

## Discussion

Analyses that utilize metabolite ratios are not a new concept in omics investigations, particularly for applications that involve medical MRI spectroscopy-generated imaging data. Notwithstanding, an unbiased strategy and application of pixel-by-pixel imaging of targeted and untargeted metabolite ratios for MSI data interpretation has been lacking. We demonstrate that ratio imaging adds a powerful new tool to mitigate sample preparation artefacts, to spatially metabotype tissue microenvironments, and to reveal spatially distinct functional variations in enzymatic and metabolic pathway activities.

Metabolite ratio imaging can uncover previously unrecognized biology, as exemplified here for the pixel-by-pixel distribution and metabolic interplay among neurotransmitter-related brain metabolites. Glutamate, glutamine, aspartate, histidine, homocarnosine and GABA are all well-recognized brain metabolites that contribute to fundamental brain activities, including synaptic transmission and energy production. Their ratios are tightly regulated for maintaining proper brain function and vary across different regions of the brain and under different physiological conditions. Disturbance of these ratios can be deleterious to neurotransmission and overall brain health. Imaging of ratios among these and all other metabolites engenders a novel mapping tool that may point to localized segmentation of metabolic activities not recognized by imaging individual metabolites. Considering ratios as proxies of enzyme reaction rates between substrate and product, ratio-derived ROIs along with their ratio and non-ratio abundances offer a novel strategy for study of disease pathophysiology, biomarker discovery and identification of new therapeutic targets (*Figures 5–7*).

As described, untargeted pixel-by pixel ratios can be derived from all detected mass spectral features, irrespective of whether the features have known structural identities. Combined with non-ratio feature data, information-rich metabolite ratio datasets offer great potential for association with other imaging or spatial omics data. Notably, generation of metabolite ratio datasets can be performed using the described R-code algorithm and applied pixel-by-pixel to individual metabolite features that have been incorporated into an ExcelL.csv format file with spot ID and pixel axis positional information. Furthermore, R codes for generating pixel data directly from.imzml files provide a convenient way to perform spatial ratio analysis for all types of MSI experiments. For pixel data with ≤10 µm raster width, single cell spatial imaging and omics data can ideally be associated with untargeted metabolite/feature ratio data to achieve single cell multiomics metadata integration. For this purpose, a P-gain can be applied to filter out significantly correlated associations, as previously described for some well-established GWAS and MWAS association studies (*Suhre et al., 2011*; *Illig et al., 2010*; *Gieger et al., 2008*; *Petersen et al., 2012*). A potential challenge for single cell metabolomics spatial data interpretation lies in possible concerns regarding metabolite data co-registration with anatomical or other single cell omics datasets. Notably, when datasets are obtained from serial sections, a mere 10 µm shift could result in cell layer disparities.

Combining ratio and non-ratio pixel data offers a powerful new tool to survey significant but unappreciated global correlations between metabolites ratios (both structurally defined and undefined) in distinct ROIs that are revealed by metabolite ratio tissue segmentation. Among other unexpected findings, we demonstrate that an elevated glutamine to glutamate ratio is observed in COX10 KO brain compared to WT within the NAc and that this ratio positively correlates with ferrous iron levels and Nat8L enzymatic activity (inferred from NAA to aspartate ratios, *Table 2*, *Figure 1—figure supplement 3*, *Figure 8—figure supplement 3*). These associations were previously unrecognized but are consistent with a whole-body feed-forward action of mitochondrial myopathy on levels of brain leptin, corticosteroids, and neurotransmitter signaling (*Southwell et al., 2023*).

Applying targeted and untargeted metabolite ratios for PCA and UMAP tissue segmentation generates novel and artefact-free ROIs linking proxy metabolic activity. One limitation for MSI ratio data interpretation is that the coverage of known metabolite structures and pathways is considerably less (typically 200–300 structurally identified species) than for LC/MS analyses. It is important to note that combined ratio and non-ratio data can also be used for segmentation. However, segmentation results using combined ratio and non-ratio data may be confounded by artefact-prone non-ratio metabolite data and thus not as clean as those using ratio alone. We speculate that combining metabolite ratios from different metabolic pathways may segment tissue with ROIs more representative of real metabotypes.

In summary, we anticipate targeted and untargeted metabolite ratio imaging to provide a powerful add-on tool for MSI experiments, revealing otherwise hidden information in acquired datasets. These ratios may serve as potential new biomarkers to distinguish anatomically distinct metabolic tissue regions and uncover otherwise unrecognized differences in either physiological cell function or from differing cellular responses to drug treatments and disease states. This ratio imaging strategy can be further extended to image pixel-by-pixel fractional abundance of isotopologues for targeted and untargeted MALDI-MSI stable isotope tracing. When coupled with the FT-ICR high resolution mass spectrometer, simultaneous imaging of isotopologue incorporation is possible for tracing molecules containing mixed stable isotope elements (e.g. $^{13}C$, $^{15}N$, $^{2}H$, $^{18}O$, $^{34}S$).

## Acknowledgements

We are thankful for Drs. Tal Nuriel, Eileen Ruth Torres, Li Gan for sharing mouse brain images used in this paper. NIH support by 1S10OD023652 (SG) for WCM's Ultra-high Resolution MALDI-FT-ICR Imaging Core Facility. NIH 5 R01 AR076029-03A1 (QC and MD), R21 NS118233-01A1 (QC and MD), R21 ES032347(QC).

## Additional information

### Competing interests

Joshua L Fischer: Joshua L Fischer is a Bruker employee. The other authors declare that no competing interests exist.

### Funding

| Funder | Grant reference number | Author |
|---|---|---|
| National Institute of Arthritis and Musculoskeletal and Skin Diseases | R01 AR076029 | Marilena D'Aurelio Qiuying Chen |
| Nation Institute | S10OD023652 | Steven S Gross |
| NIH | R21 NS118233-01A1 | Qiuying Chen Marilena D'Aurelio |
| NIH | R21 ES032347 | Qiuying Chen |

The funders had no role in study design, data collection and interpretation, or the decision to submit the work for publication.

### Author contributions

Huiyong Cheng, Conceptualization, Data curation, Formal analysis, Methodology, Writing – original draft; Dawson Miller, Nneka Southwell, Paola Porcari, Joshua L Fischer, Isobel Taylor, J Michael Salbaum, Claudia Kappen, Fenghua Hu, Cha Yang, Resources, Methodology; Kayvan R Keshari, Marilena D'Aurelio, Conceptualization, Resources, Methodology; Steven S Gross, Conceptualization, Investigation, Methodology, Writing – review and editing; Qiuying Chen, Conceptualization, Resources, Data curation, Software, Formal analysis, Supervision, Funding acquisition, Validation, Investigation, Visualization, Methodology, Writing – original draft, Project administration, Writing – review and editing

## Author ORCIDs
Fenghua Hu https://orcid.org/0000-0002-6447-9992
Qiuying Chen https://orcid.org/0000-0001-5909-3959

Reviewer #2 (Public review): https://doi.org/10.7554/eLife.96892.3.sa1
Author response https://doi.org/10.7554/eLife.96892.3.sa2

## Additional files

### Supplementary files
Supplementary file 1. Comparison of untargeted ratio imaging R workflow to other ratio software.

Supplementary file 2. Brain glutamine to glutamate ratio negatively correlates FeCl2 using combined ratio and non-ratio pixel data and spearman correlation.

Supplementary file 3. List of glutamate metabolism related metabolites for ratio generation and PCA segmentation of brain sections.

Supplementary file 4. List of lyso PEs for ratio generation and PCA segmentation of brain sections.

Supplementary file 5. Comparing correlation of neurotransmitter related metabolites and ratios in ROI7 of WT and KO brains.

MDAR checklist

### Data availability
Source data files have been provided for Figures.

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
