## [Editor Report · eLife Assessment]

This **valuable** study describes a software package in R for visualizing metabolite ratio pairs. The evidence supporting the claims of the authors is **solid** and broadly supports the authors' conclusions. This work would be of interest to the mass spectrometry community.

---

## [Referee Report · Reviewer #2 (Public review)]

Summary:

In the article, the authors describe their software package in R for visualizing metabolite ratio pairs. I think the work would be of interest to the mass spectrometry community.

Strengths:

The authors describe a software that would be of use to those performing MALDI MSI. This software would certainly add to the understanding metabolomics data and enhance the identification of critical metabolites.

Weaknesses:

The figures are difficult to interpret/ analyze in their current state but are significantly better in the revision.

---

## [Author Response]

The following is the authors’ response to the original reviews.

**Public Reviews:**

**Reviewer #1 (Public Review):**
Cheng et al explore the utility of analyte ratios instead of relative abundance alone for biological interpretation of tissue in a MALDI MSI workflow. Utilizing the ratio of metabolites and lipids that have complimentary value in metabolic pathways, they show the ratio as a heat map which enhances the understanding of how multiple analytes relate to each other spatially. Normally, this is done by projecting each analyte as a unique color but using a ratio can help clarify visualization and add to biological interpretability. However, existing tools to perform this task are available in open-source repositories, and fundamental limitations inherent to MALDI MSI need to be made clear to the reader. The study lacks rigor and controls, i.e. without quantitative data from a variety of standards (internal isotopic or tissue mimetic models for example), the potential delta in ionization efficiencies of different species subtracts from the utility of pathway analysis using metabolite ratios.

We thank the reviewer for comments on the availability of four other commercial and open-source tools for performing ratio imaging: ENVI Geospatial Analysis Software, MATLAB image processing toolbox, Spectral Python (SPy) and QGIS. We now highlight these in the introduction (page 3 line 80-86). However, in contrast to these target ratio imaging methods, our approach uniquely enables the untargeted discovery of correlated (or anti-correlated) ratios of molecular features, whether the species are structurally known or unknown.

ENVI Geospatial Analysis Software and MATLAB image processing toolbox for hyperspectral imaging are both paid programs, limiting free access and software evaluation for the potential application of untargeted ratio-metric imaging. We are able to evaluate the application of MATLAB RatioImage since Weill Cornell Medicine has an institutional subscription for Mathwork-MATLAB. Notably, MATLAB RatioImage computes and displays an individual intensity modulated ratiometric image by choosing a numerator and denominator image. This software tool only images the ratios of selected metabolites from an input list of multiple species and does not allow for the possibility of untargeted ratiometric images of all metabolite pairs.

While Spectral Python (SPy) and QGIS are both freely-available software packages, and both can perform individual metabolite ratio images, neither allows for untargeted ratiometric imaging of all pairs from a multiple metabolite input list. Table S1 (below) provides a comparison of the ratio imaging tool that we offer in comparison with other previously available tools.

We appreciate the reviewer’s insightful comments on differential ionization efficiency among metabolites and the importance of using stable isotope internal standard to gain absolute quantification.

A fundamental advantage of our ratiometric imaging tool is to provide better image contrast for tissue regions with differential ionization efficiency, with the potential to discover new “metabolic” regions that can be revealed by metabolite ratio. Note that comparison for ratio image abundance is limited to tissue groups in the equivalent region which is expected to have similar ionization efficiency for given metabolites. Furthermore, the power of our strategy is to provide untargeted (and targeted) ratio imaging as a hypothesis generation tool and this use does not require absolute quantification. If cost was not an issue, an extensive group of stable isotope standards could theoretically be used for absolute metabolite quantification of target metabolites with known identity.

Using the tissue mimetic model, we generate calibration curve for stable isotope standards spiked in carboxymethylcellulose (CMC)-embedded brain homogenate cryosections and quantify the concentration of brain glucose, lactate and ascorbate concentrations. Similar ratio images among these metabolites are obtained from abundance data compared to quantified concentration data (Fig S3). While stable isotope standards are often used to obtain quantitative concentration of metabolite/lipid of interest, it is not applicable for untargeted metabolite ratios that include an assessment of structurally undefined species. Nevertheless, our data indicates that absolute quantification is not necessary for the targeted and untargeted ratio imaging described here (Page 6, line 196-205).

**Reviewer #2 (Public Review):**
Summary:In the article, "Untargeted Pixel-by-Pixel Imaging of Metabolite Ratio Pairs as a Novel Tool for Biomedical Discovery in Mass Spectrometry Imaging" the authors describe their software package in R for visualizing metabolite ratio pairs. I think the novelty of this manuscript is overstated and there are several notable issues with the figures that prevent detailed assessment but the work would be of interest to the mass spectrometry community.Strengths:The authors describe a software that would be of use to those performing MALDI MSI. This software would certainly add to the understanding of metabolomics data and enhance the identification of critical metabolites.Weaknesses:The authors are missing several references and discussion points, particularly about SIMS MSI, where ratio imaging has been previously performed.There are several misleading sentences about the novelty of the approach and the limitations of metabolite imaging.Several sentences lack rigor and are not quantitative enough.The figures are difficult to interpret/ analyze in their current state and lack some critical components, including labels and scale bars.

We thank reviewer for very helpful comments. The tone of the manuscript has been adjusted to highlight the real novelty of this method in the ease of computing and application to MS specific projects (abstract line 26-30). All figures have been updated to include labels and scale bars with improved resolution. References for ratio imaging use of SIMS MSI has been added in the introduction (Page 3, line 80-89).

**Recommendations for the authors:**

**Reviewer #1 (Recommendations For The Authors):**
Major Comments:In the Abstract it is stated that: "the research community lacks a discovery tool that images all metabolite abundance ratio pairs." However, the following tools exist that perform this fundamental task.A "pixel by pixel" data frame in .csv form has a very similar data structure to many instruments like satellite imaging or other hyperspectral tools. It is true this does not exist in the MALDI-specific context, but it would not be difficult to perform this task on the following programs. Highlight the novelty here is not ratios but the ease of computing them and the application in the specific project. Also, describe the available tools and what shortcomings others lack that this package provides. A supplemental table of MSI data analysis tools and the function of each would be a good addition.List of tools to perform band ratio computation with minimal modification:(1) ENVI IDL: geospatial imaging tool that allows ratio computation between spectral bands.(2) MATLAB image processing toolbox for hyperspectral imaging.(3) Spectral Python package (SPy).(4) QGIS with plugins can be used for hyperspectral image analysis with a ratio between bands.

We revised the abstract and introduction to include novelty and comparison to other existing methods listed in Table S1.

"untargeted R package workflow" - If there are functions used outside the SCiLS Lab API client then write it up and include a GitHub link for open access to fit the mission of eLife.

As shown in Scheme I. We develop two types of codes for untargeted ratio imaging. The first type uses Scils lab API client to extend the function of targeted and targeted ratio imaging and all related spatial image analysis. This is suitable for Scils lab users. The second type does not require Scils lab API, it allows extracting pixel data from imzml file then proceed targeted and untargeted imaging and analysis. Both codes are now deposit in Github via public access (https://github.com/qic2005/Untargeted-massspectrometry-ratio-imaging.git).

"across cells and tissue subregions" The value in reporting cell type and tissue type-specific differences in any metric is powerful, but not done in this paper. Only whole samples are compared such as "KO vs WT" and the annotations in Figure 3 are not leveraged for increased biological relevance. This paper treats each image as a homogenization experiment in a practical sense beyond just visually inspecting each image. Remove this claim or do the calculations on region/tissue/cell-type specific differences with the appropriate tools to show the data beyond simple heat map images.

We have deleted the sentence containing across cells and tissue subregions from the abstract.

"enhances spatial image resolution" Clarify. The resolution in MALDI is set by the raster size of the pixels which is an instrument parameter and cannot be changed post-acquisition. Image-specific methods to increase resolution exist, but dividing the value in one peak column by another does not change functional resolution in the context of the instruments here.

We thank reviewer for pointing out this typo. We have changed it to enhance spatial image contrast in the abstract (line 34).

"pixel-by-pixel imaging of the ratio of an enzyme's substrate to its derived product offers an opportunity to view the distribution of functional activity for a given metabolic pathway across tissue" - Appropriately calibrate the impact of this work and correct this statement to better reflect the capabilities of this approach. Do not oversell the exploration of pathway activity since the raw quantity reported as relative abundance does not provide biologically interpretable pathway information. This is due to unaccounted differences in ionization efficiencies between analytes in a pathway and lack of determination of rate. Without a calibration curve and more techniques on the analytical chemistry side of the project, it is possible a relative abundance of one analyte (like the product of a pathway) could be higher than the relative abundance of another analyte (a precursor), but due to structural differences, the actual quantity of the higher relative abundance species could be significantly different or even lower than its counterpart. Secondly, "functional activity" cannot be assessed in this manner without isotopic labeling or additional techniques. This does not subtract from the overall validity and impact of the work, but highlighting these shortcomings and slight alterations to the claim are important for a multidisciplinary audience.

Although we show that abundance ratio results in similar image to concentration ratio for brain metabolites such as lactate, glucose and ascorbate, we agree with the reviewer that abundance ratio is different from the absolute concentration ratio in numerical value due to difference in ionization efficiency. We delete the sentence “pixel-by-pixel imaging of the ratio of an enzyme's substrate to its derived product offers an opportunity to view the distribution of functional activity for a given metabolic pathway across tissue" from the abstract. We apologize for not clarifying this application more clearly. We meant to compare pathway activity among the equivalent and similar pixel/regions of tissues from different biological groups, given the assumption that ionization efficiency is identical for equivalent pixel from different tissue sections (i.e. same cell type and microenvironment), especially for metabolites with similar functional structure in the same pathway. For example, fatty acids with different chain length and phospholipid with same head groups are expected to have similar ionization efficiency in the same tissue pixel/region. We have thereby rewritten this section (Page 7, line 239-247).

"We further show that ratio imaging minimizes systematic variations in MSI data by sample handling and instrument drift, improves image resolution, enables anatomical mapping of metabotype heterogeneity, facilitates biomarker discovery, and reveals new spatially resolved tissue regions of interest (ROIs) that are metabolically distinct but otherwise unrecognized."

Instrument drift is not accounted for by ratios as it impacts the process before ratio computation. "metabotype" - spelling?

Instrument drift here refers to individual ion abundance changes during long data acquisition. Ratio may offer a better read-out than individual metabolite abundance alone. However, for acquired data after total ion normalization, ratio data would not have difference from non-ratio data. Therefore, we delete instrument drift from the sentence (Page 2, line 33, and Page 3, line 99)

Metabotype is a term widely used for metabolomics field. It is categorized by similar metabolic profiles, which are based on combinations of specific metabolites. https://nutritionandmetabolism.biomedcentral.com/articles/10.1186/s12986-020-00499-z

Results 3: Justify the claim that the ratio reduces artifacts. A ratio is the value from one m/z area over another and would seem that the quality of the ratio would be always lower than the individually higher quality pixel signal of the two analytes that compose a ratio.

Ratio images are indeed the heatmaps of pixel-by-pixel ratio data, set by the scale of all ratio values. For very abundant ion pairs, their individual image may not be better than the ratio image, depending on the abundance changes among pixels within tissue sections. Similarly, the quality of ratio image may not be higher than the individual image if distribution of ratios does not change much among pixels in tissue sections. For example, metabolite or lipids in Figures 2 and 5 are abundant, but non-ratio images do not have better quality than ratio images. Furthermore, ratio image provides additional information on how the ratio of the two metabolite pair changes pixel-by pixel in all tissue sections, such additional information could be useful for data interpretation.

Results 4: The metabolite pairs are biologically sensible but should be clearly stated that they do not account for differences in ionization efficiency between metabolites and cannot provide quantitative pathway analysis with a high degree of biological confidence.

We apologize for not clarifying this application more clearly. We meant to compare pathway activity among the equivalent and similar pixel/regions of tissues from different biological groups, given the assumption that ionization efficiency is identical for equivalent pixel from different tissue sections (i.e. same cell type and microenvironment), especially for metabolites with similar functional structure in the same pathway. For example, fatty acids with different chain length and phospholipid with same head groups are expected to have similar ionization efficiency in the same tissue pixel/region. We have thereby rewritten this section (Page 7, 239-247, 254-255).

Results 4: "cell-type specific metabolic activity at cellular (10 µm) spatial resolution" Prove the cell type differences with IHC coregistration or MALDI IHC if you want to make claims about them. Just visually determining a tissue type of a scan of a slide is inadequate to support this claim.

We agree with reviewer’s comments. We meant to provide additional information on cellular level metabolic activity such as adenosine nucleotide phosphorylation status (ATP/AMP) ratio at 10µm resolution. Hippocampus neurons provide a good example for depicting this utility. We have rewritten the claim to highlight the role of ratio imaging in providing additional metabolic information (Page 8, line 288-290).

Minor Comments:Table 2 "Aspartiate" spelling

We have corrected it.

Describe the process and mathematical background for ratio computation in the Methods section. As this paper introduces a package, describing its underlying functions has value.

We have added R-script comments to illustrate the untargeted ratio calculation using the R-mathematical function of combination and division between any two metabolite pairs in a data matrix (Page 4, line 139-141)

"we annotate missing values with 1/5 the minimum value quantified in all pixels in which it was detected" This is explicit ie only values with exactly 1/5 the value are annotated" - make it clear this is a threshold.

We apologize for misunderstanding. Missing values are either have no value or have solid zero in their abundance. We first calculate the minimum abundance of a particular m/z among all pixels with detectable abundance (i.e. excluding non-missing values), then use 1/5 this minimum value as a threshold to annotate missing value (Page 4, 133-139).

Figure 1: legend scils is branded SCiLS and EXCEL does not need caps lock (Excel).

Figure 1 legend has been corrected.

Conflicts of interest "None" - there are Bruker employees on a paper about MALDI method development in a field they dominate.

We added Joshua Fischer as a Bruker employee.

Figure 3: The legend does not describe the purple arrow in J.

Purple arrow description is added to figure legend.

Figure 5: Fix orientation inconsistencies in G, H, I, and J. Especially in J - they are opposite directions. This is arbitrary and determined in SCiLS lab with simple rotation.

Orientation has been made consistent in G,H, I and J.

Figure S8: Provide exact number of biological and technical replicates used to generate this figure.

Figure S8, now Figure S9, was generated from 4 biological replicates of KO and 4 biological replicates of WT brain section in the ROI7 region. This information has been added to the figure legend.

Figure S9: Make consistent orientation of all brains

We have made brain orientations consistent.

In addition to ionization efficiencies impacting the value of the numeric relative abundance where ratio computation originates from, it should be mentioned how different classes of metabolites are differentially impacted by the euthanasia and collection methods used for various tissue types. For example, it is well established the ATP/AMP ratio can change drastically from tissue collection.

We have added this to page 8, line 315-319.

Perform standards to adjust for ionization efficiency between different m/z features.

Untargeted ratio imaging serves as an add-on MSI data analysis tool with primary use in comparing ratio among equivalent regions/pixels with similar ionization efficiencies. It is a hypothesis generation tool. Standards adjust for ionization efficiency would be a great idea for a more accurate assessment of ratio values. Due to the cost and availability of stable isotope standards for different m/z, we chose glucose, lactate and ascorbate to showcase that abundance ratio and concentration ratio result in similar images among example brain metabolite lactate, glucose and ascorbate (page 6, 196-205).

Add more controls to support the claims.

We have 4 biological replicates for each genotype of brain. We have added the number of controls in all figure legends.

Significantly tone down the claims, it is unclear how knowledgeable the authors are about the current literature of SW regarding MALDI.

The tone has been significantly tuned down throughout the revised manuscript.

**Reviewer #2 (Recommendations For The Authors):**
Abstract:"relative abundance of structurally identified and yet-undefined metabolites across tissue cryosections" is misleading, since tandem MS can be performed in an imaging context and is often also compatible with the same instrument.

We have deleted this sentence in the abstract.

Intro:Paragraph 1: The authors mention MALDI and DESI, but I would argue that SIMS is more abundantly used than DESI within single-cell applications.

We have added SIMS to the introduction Page 3, line 67.

Paragraph 2: While it may not be all detected pairs, there are many examples of ratio imaging in the MALDI MSI and SIMS communities, particularly for bacterial signaling. These would be important examples to reference.

We have added the application of SIMS ratio imaging to the introduction, page 3, line 74-75.

Materials :Paragraph 1: More specificity on sample size is required. 3 or 4 per group is not specific. Which has four and which has three? Why are they different?

We have corrected sample numbers for specific genotype in the text and figure legends. The number of sections per group is different due to the availability of fresh-frozen tissues (Page 4, line 115-117).

Results:Paragraph 1: Am I correct in reading that an .imzml can't be used directly? Why not?

Imaging Mass Spectrometry Markup Language (imzml) is a common data format for mass spectrometry imaging. It was developed to allow the flexible and efficient exchange of large MS imaging data between different instruments and data analysis software (Schramm et al, 2012). It contains two sets of data: the mass spectral data which is stored in a binary file (.ibd file) to ensure efficient storage and the XML metadata (.imzml file) which stores instrumental parameters, sample details. Therefore, it can’t be used directly. We have added this to result 1(Page 5, line 160-169).

Paragraph 4: "Additionally, nonlipid small molecule metabolites suffer from smearing and/or diffusion during cryosection processing, including over the course of matrix deposition for MALDI-MSI." This is misleading. There are several examples of MALDI MSI of small metabolites that are nonlipids, where smearing or diffusion have not occurred. It would be beneficial to have a more accurate discussion of this instead. The authors should also provide some evidence of this, since they continue to focus on it for the full paragraph and don't provide references.

We initially meant the poor image quality of small molecule metabolites is due to its interaction with aqueous phase of spraying solution, rapid degradation rate and matrix interference. We have deleted this sentence in the revised version.

Section 5 Paragraph 2; "However, ratio imaging revealed a much greater aspartate to glutamate ratio in an unusual "moon arc" region across the amygdala and hypothalamus relative to the rest of the coronal brain." Much greater isn't scientifically accurate or descript. Use real numbers and be quantitative.

We used pixel data from all 8 sections to obtain quantitative changes in the ratio-generated “moon arc” region compared to the rest of coronal brain (page 8, line 331-337). Ratio imaging revealed a average of 1.59-fold increase in aspartate to glutamate ratio in an unusual “moon arc” region across the amygdala and hypothalamus (mean abundance 0.563 in 6345 pixels) relative to the rest of the coronal brain (mean abundance 0.353 in 45742 pixels, Figure 5D). Similar but different arc-like structures are encompassed within the ventral thalamus and hypothalamus, wherein glutamate to glutamine ratio show a 1.63-fold increase in intensity compared to the rest of the brain (mean abundance of 0.695 in 7108 pixels vs 0.428 in 44979 pixels, Figure 5E).

Section 8 Paragraph 2: "UMAPing" is not scientifically written.

We have replaced UMAPing with UMAP.

Figure 2 is difficult to interpret, given the small sizes of the images. Align the images, reduce the white space, clearly label the different tissues, add scale bars, increase size, etc. This applies to all figures, except for 3. This will make it possible to review.

All figures have been resized by removing extra space between sections.

Figure 3. There seems to be a change in tissue after section I, so a different diagram would be helpful. SCD has a high abundance in an area that seems to be off of the tissue. Can the authors explain this? Some of the images also appear to be low signal-to-noise. Example spectra in the SI would be helpful, so I can more accurately judge the quality of the data.

We apologize for the discrepancy. All images are from the same sample. We initially cropped the individual image from multiple page PDF plot, then inserted it in Figure 3. Resizing and cropping inconsistency may lead to the small difference in image size. In the revised version, we plot all images in one page, which eliminates the inconsistency.

Figure 3 example pixel data, ratio pixel data, mass spectra and ratio images can be downloaded below:

https://wcm.box.com/s/2d5jch45ar8upjzytljnylt6doewcsqc